# Improvements to the representation of BVOC chemistry-climate interactions in UKCA (vn11.5) with the CRI-Strat 2 mechanism: Incorporation and Evaluation

James Weber[1], Scott Archer-Nicholls[1], Nathan Luke Abraham[1,2], Youngsub M. Shin[1], Thomas J. Bannan[3], Carl J. Percival[4], Asan Bacak[5], Paulo Artaxo[6], Michael Jenkin[7], M. Anwar H. Khan[8], Dudley E. Shallcross[8], Rebecca H. Schwantes[9,10], Jonathan Williams[11,12], Alex T. Archibald[1,2]

*Correspondence to*: James Weber (jmw240@cam.ac.uk)

[1]Centre for Atmospheric Science, Department of Chemistry, University of Cambridge, Cambridge, CB2 1EW, UK

[2] National Centre for Atmospheric Science, Department of Chemistry, University of Cambridge, CB2 1EW, UK

[3]School of Earth and Environmental Sciences, University of Manchester, Manchester, M13 9PL, UK

[4]NASA Jet Propulsion Laboratory, California Institute of Technology, 4800 Oak Grove Drive, Pasadena, CA 91109, USA.

[5]Turkish Accelerator & Radiation Laboratory, Ankara University Institute of Accelerator Technologies, Gölbaşi Campus, 06830 Gölbaşi, Ankara, Turkey.

[6]Physics Institute, University of São Paulo, Rua do Matão 1371, CEP 05351-015, São Paulo, Brazil

[7]Atmospheric Chemistry Services, Okehampton, Devon, EX20 4BQ, UK

[8]Biogeochemistry Research Centre, School of Chemistry, University of Bristol, Cantock's Close, Bristol, BS8 1TS, UK

[9]Chemical Sciences Laboratory, National Oceanic and Atmospheric Administration, Boulder, CO 80305, USA

[10]Cooperative Institute for Research in Environmental Sciences, University of Colorado, Boulder, CO, 80309, USA

[11]Department of Atmospheric Chemistry, Max Planck Institute for Chemistry, Mainz, 55128, Germany.

[12]Energy, Environment and Water Research Center, The Cyprus Institute, Nicosia, Cyprus

**Abstract** We present the first incorporation of the Common Representative Intermediates version 2.2 tropospheric chemistry mechanism, CRI v2.2, combined with stratospheric chemistry, into the global chemistry-climate United Kingdom Chemistry and Aerosols (UKCA) model to give the CRI-Strat 2 mechanism. A rigorous comparison of CRI-Strat 2 with the earlier version, CRI-Strat, is performed in UKCA in addition to an evaluation of three mechanisms, CRI-Strat 2, CRI-Strat and the standard UKCA chemical mechanism, StratTrop vn1.0, against a wide array of surface and airborne chemical data.

CRI-Strat 2 comprises a state-of-the-art isoprene scheme, optimised against the MCM v3.3.1, which includes isoprene peroxy radical isomerisation, $HO_x$-recycling through the addition of photolabile hydroperoxy aldehydes (HPALDs) and IEPOX formation. CRI-Strat 2 also features updates to several rate constants for the inorganic chemistry including the reactions of inorganic nitrogen and $O(^1D)$.

The update to the isoprene chemistry in CRI-Strat 2 increases OH over the lowest 500m in tropical forested regions by 30-50%, relative to CRI-Strat, leading to an improvement in model-observation comparisons for surface OH and isoprene relative to CRI-Strat and StratTrop. Enhanced oxidants also cause a 25% reduction in isoprene burden and an increase in oxidation fluxes of isoprene and other biogenic volatile organic compounds (BVOCs) at low altitudes with likely impacts on subsequent atmospheric lifetime, aerosol formation and climate.

By contrast, updates to the rate constants of $O(^1D)$ with its main reactants relative to CRI-Strat reduces OH in much of the free troposphere, producing a 2% increase in the methane lifetime, and increases the tropospheric ozone burden by 8%, primarily from reduced loss via $O(^1D) + H_2O$. The changes to inorganic nitrogen reaction rate constants increase the $NO_x$ burden by 4% and shift the distribution of nitrated species closer to that simulated by StratTrop.

CRI-Strat 2 is suitable for multi-decadal model integrations and the improved representation of isoprene chemistry provides an opportunity to explore the consequences of $HO_x$-recycling in the United Kingdom Earth System Model (UKESM1). This new mechanism will enable a re-evaluation of the impact of BVOCs on the chemical composition of the atmosphere and probe further the feedback between the biosphere and the climate.

## 1. Introduction

Isoprene (2-methyl-1,3-butadiene) makes up 70 % of all non-methane BVOC emissions with annual average emissions of 594 ± 34 Tg/year over the period 1980-2010 (Sindelarova et al., 2014). Isoprene's rapid chemical oxidation in the atmosphere by OH, $O_3$ and $NO_3$ directly affects the tropospheric oxidising capacity, ozone burden and the processing of other trace gases like methane (e.g. Archibald et al, 2011) while also serving as an important source of secondary organic aerosol (SOA) (e.g., Kelly et al., 2018). Thus, isoprene has substantial effects on the radiative balance of the atmosphere, both directly via production of SOA and ozone, and indirectly via its changes to the oxidising capacity of the atmosphere influencing methane lifetime and production of other aerosol species such as from oxidation of monoterpenes and $SO_2$. An accurate representation of isoprene's chemical behaviour in climate models is essential to understanding the feedbacks between the biosphere and the rest of the Earth system and thus capturing isoprene's climatic impact.

However, the treatment of isoprene in the chemistry schemes of many climate models is outdated or oversimplified (e.g. Squire et al., 2015). The last decade has seen significant advances in our understanding of the isoprene oxidation pathway, principally the concept of rapid, intramolecular hydrogen shifts (H-shifts), also termed isomerisation reactions, in the isoprene hydroxy peroxy radicals (frequently termed ISOPOO). Predictions from theoretical work (Peeters et al., 2009, Peeters et al., 2014) and observations (Crounse et al., 2011; Teng et al., 2017; Wennberg et al., 2018) have established this pathway to be competitive with the traditional bimolecular reactions of the peroxy radical with NO, $NO_3$, $HO_2$ and $RO_2$ in certain conditions such as low $NO_x$(=NO + $NO_2$) environments. These H-shifts reactions lead to the production of $HO_x$(=OH + $HO_2$) either directly or indirectly following the degradation of the isomerisation products (e.g., Archibald et al., 2010, Jenkin et al., 2015, Wennberg et al., 2018).

This process, termed $HO_x$-recycling, has been shown to be important for low-$NO_x$, high-isoprene regions of the atmosphere (Butler et al., 2008, Lelieveld et al., 2008). By adding a simple, fixed yield OH production pathway from ISOPOO to represent OH production from hydroperoxy aldehydes (HPALDs), Archibald et al (2011) simulated an 8-18% increase in tropospheric $O_3$ burden while the tropospheric OH burden increased by 17% in the present day (PD) and by 50% in a pre-industrial (PI) atmosphere featuring 1860 emissions of key chemical species such as $NO_x$, CO and isoprene. Consequently, the lifetime of methane was predicted to decrease between 11% (in a future climate scenario) and 35% (in the PI). This illustrated the significant impact that such a process could have on our understanding of the PI atmosphere (and the radiatively active components therein), and thus the PD-PI change and climate sensitivity. While the greatest change to the chemistry was simulated in the boundary layer (BL), convection of isoprene and its oxidation products into the free troposphere resulted in this added chemistry having global impacts. The effect on oxidants from $HO_x$-recycling influences the lifetimes of isoprene and other BVOCs such as monoterpenes and thus the extent of their dispersion and the location of the subsequent SOA formation. Karset et al. (2018) found that when lower oxidant fields were applied to the PI atmosphere isoprene, monoterpenes, $SO_2$ and other key aerosol precursors were more dispersed from their sources, reaching higher altitudes and enhancing particle number concentration in the remote free troposphere. The radiative impact of the resulting aerosols was greater due to their enhanced lifetime (from slower deposition) and the highly non-linear relationship between aerosol number and cloud forcing where the addition of a given concentration of aerosol has a much greater impact in remote regions where the background concentration of aerosol is smaller (Chen et al., 2016). The importance of oxidants to BVOCs and aerosol was also shown in Sporre et al (2020) where models with an interactive oxidant scheme simulated a BVOC-driven depletion of oxidants and attendant greater dispersion of BVOCs and their oxidation products (including SOA precursors). In contrast, a prescribed oxidant approach saw BVOC oxidation confined far more to source regions, reducing dispersion.

Change to oxidant fields also perturb the oxidation pathways of $SO_2$. In the United Kingdom Chemistry and Aerosols (UKCA) model, $SO_2$ can be oxidised in the gas phase by OH to yield $H_2SO_4$ or in the aqueous phase by $O_3$ or $H_2O_2$ (Mulcahy et al., 2020). This has consequences for the aerosol mass and number distributions because only $H_2SO_4$ can nucleate new particles in UKCA, therefore amplifying the gas phase pathway over the aqueous pathways leads to a greater number of smaller aerosols. Thus, uneven changes to these pathways can alter the size and number distribution of the aerosol population, affecting the radiative properties of aerosols and clouds. Decreases in OH in other UKCA studies (Weber et al., 2020a, O'Connor et al., 2020) have resulted in simulated reductions in particle number concentration and cloud droplet number concentration. The resulting negative cloud radiative forcing is smaller in magnitude as the lower cloud droplet number concentration (CDNC) makes the clouds less "bright" (Twomey et al., 1974). The impact of different oxidant schemes on the burden and lifetime of DMS, an important $SO_2$ precursor, and the impact to sulphate aerosol transport is highlighted by Mulcahy et al. (2020).

While Archibald et al (2011) used a relatively simple approach to simulate $HO_x$-recycling, further advances in the chemical understanding have led to a near explicit representation of $HO_x$-recycling being incorporated into comprehensive mechanisms including the Master Chemical Mechanism (MCM v3.3.1) (Jenkin et al., 2015) and the

CalTech isoprene scheme (Wennberg et al., 2018). However, such mechanisms are far too large for use in global chemistry-climate models.

There exist a few reduced mechanisms featuring this state-of-the-art isoprene chemistry suitable for use in chemistry-climate models including the CalTech reduced isoprene scheme (Bates et al., 2019) and the Common Representative Intermediates mechanism v2.2 (CRI v2.2) (Jenkin et al., 2019), the focus of this work. The CRI v2.2 is an update to the Common Representative Intermediate v2.1 mechanism (Jenkin et al., 2008, Utembe et al., 2009, Watson et al., 2008) and was developed from the fully explicit Master Chemical Mechanism (MCM) version 3.3.1 (Jenkin et al., 2015) which describes the degradation of organic compounds in the troposphere. In the CRI framework, species are lumped together into surrogate molecules whose behaviour is optimised against the fully explicit MCM. A description of CRI v2.2 is given in Jenkin et al. (2019). The CRI v2.1, along with the corresponding stratospheric chemistry, has already been incorporated into UKCA as CRI-Start (CS) (Archer-Nicholls et al., 2020) as an alternative to the simpler but more widely used STRAT-TROP (ST) chemistry scheme (Archibald et al., 2020a), the scheme used for UKESM's contributions to CMIP6 (e.g. Sellar et al., 2020, Thornhill et al., 2020).

Using the reduced Caltech Isoprene Mechanism, which includes H-shifts of ISOPOO in GEOS-CHEM, Bates et al (2019) simulated significant increases in OH (>100%) and $HO_2$ (up to 50%) over the Amazon and other forested tropical regions as a result of the $HO_x$-recycling. After implementing updated rate constants for isoprene H-shifts in GEOS-CHEM Møller et al. (2019) also found that globally around 30% of all isoprene peroxy radicals undergo at least one H-shift reaction resulting in an OH yield of 47% per isoprene molecule and that adding all isoprene H-shift reactions increased boundary layer OH by up to a factor of three in the Amazon. Using CESM/CAM-CHEM and the MOZART-TS2 mechanism, Schwantes et al (2020) showed reasonable agreement for some isoprene oxidation products over the Southeast USA.

Jenkin et al. (2019), using CRI v2.2 in the STOCHEM Lagrangian chemistry-transport model, showed the significant influence of $HO_x$ recycling in CRI v2.2 simulating a 6.4% increase in the tropospheric OH burden relative to the CRI v2.1 and increases of surface OH of 20-50% over much of the forested tropical regions. Khan et al. (2021), using the same setup, also simulated enhanced surface OH and attendant decreases in methane lifetime (0.5 years) and isoprene burden (17%).

However, while the reduced mechanisms featuring $HO_x$-recycling chemistry have been tested in chemistry-climate models, less work has been done in terms of multi-species comparison to observations and detailed analysis of the effect to global atmospheric composition. This study introduces the CS2, based on CRIv2.2 and expanded with stratospheric chemistry, as a mechanism in UKCA, evaluates its performance against observational data, and compares its output and key processes to the related CS mechanism and well-established ST mechanism. By providing a wide-ranging comparison to observations and a detailed description of the changes CS2 causes to global and regional atmospheric chemistry, this current work builds on the existing literature to develop further our understanding of the consequences of $HO_x$-recycling.

## 2. Development of CS2 - incorporation of CRI v2.2 into UKCA

It is important to note that the CRI v2.2 mechanism, like the CRI v2.1 mechanism, is strictly a tropospheric chemistry scheme. In developing the whole atmosphere mechanism CS, Archer-Nicholls et al (2020) merged the CRI v2.1 mechanism with the Stratospheric chemistry scheme (Morgenstern et al., 2009) in UKCA (Table 1) to allow this scheme to be used within UKESM1 (Sellar et al., 2019). The same approach was taken in this work with the Stratospheric scheme unchanged and tropospheric scheme switched from CRI v2.1 to CRI v2.2. Therefore, to differentiate the "CRI v2.2" mechanism used in UKCA in this work from the solely tropospheric CRI v2.2 mechanism described on the CRI v2.2 website (http://cri.york.ac.uk/), the UKCA mechanism will henceforth be referred to as CRI-Strat 2 (CS2) (Table 1). A full description of the changes made to CS to update it to CS2 is given in the supplement Section 1.1 while a summary of the changes is now discussed.

CS2 features a significant update to isoprene oxidation chemistry relative to CS with the incorporation of 1,6 and 1,4 H-shift reactions of isoprene peroxy radicals as well as an update to the organonitrate scheme (as detailed in Jenkin et al., 2019). CS2 also features updates to multiple reaction rate constants (which were out of date in CS (Archer-Nicholls et al., 2020)) to the best of our understanding as documented in the IUPAC Task Group on Atmospheric Chemical Kinetic Data Evaluation (http://iupac.pole-ether.fr/). Changes to the rate constants of the reactions of $O(^1D)$ with $H_2O$, $O_2$ and $N_2$; rate constants of multiple inorganic nitrogen reactions such as those forming PAN-type species, $HONO_2$ and the $HO_2$+NO reaction and the rate constants of organic peroxy radicals ($RO_2$) with NO and $NO_3$. These updates ensure consistency between the CS2 mechanism incorporated in UKCA and that described on the CRI v2.2 website (http://cri.york.ac.uk/). The photolysis of glyoxal, formaldehyde and propionaldehyde was also updated (see SI Section S6).

CS2 has 9 more species than CS (Tables 1, 2) as well as 46 additional bimolecular reactions, 12 additional photolysis reactions and 8 additional uni/termolecular reactions (Table 1). This leads to a modest increase in runtime (6%) compared with CS whose runtime was already ~75% greater than ST. Incorporation of CS2 into UKCA involved extensive use of the UM-UKCA virtual machine environment (Abraham et al., 2018).

The main update to the isoprene chemistry is the inclusion of 1,6 and 1,4 H-shift reactions of the isoprene peroxy radical (termed RU14O2 in CRI nomenclature). The 1,6 H-shift process is well studied (Peeters et al., 2009, Crounse et al., 2011, Teng et al., 2017, Wennberg et al., 2018) and follows the $k_{bulk1,6H}$ rate coefficient described in Jenkin et al. (2019), capturing the dependence of isomerisation on both temperature and the rates of reaction of RU14O2 with the standard bimolecular partners (NO, $NO_3$, $HO_2$ and $RO_2$). This pathway yields hydroperoxy aldehydes (HPALDs, termed HPUCARB12 in CS2) and dihydroperoxy carbonyls peroxy radicals (DHPR12O2). The photolysis of the highly photolabile HPALD (HPUCARB12), and its product HUCARB9 (unsaturated hydroxy carbonyl), are key routes for $HO_x$ regeneration.

The production of the isoprene epoxy diol (IEPOX) from the isoprene hydroperoxide (RU14OOH) and the hydroxymethyl-methyl-a-lactone (HMML) also represent important updates (Jenkin et al., 2019). IEPOX and HMML are known SOA precursors (Nguyen et al., 2014; Nguyen et al 2015; Allan et al., 2014) and so their addition may enable a more explicit representation of SOA formation within the CRI framework, as opposed to the current framework whereby SOA formation is represented by the condensation on existing aerosol of a single inert tracer, Sec_Org, which is made from monoterpene oxidation (Mann et al., 2010; Mulcahy et al., 2020). This is beyond the scope of this paper but will be a focus of future work.

The introduction of HPUCARB12 and HUCARB9 necessitates a careful update to the FASTJX photolysis scheme used by UKCA (Telford et al., 2013). The cross-sectional dependence of wavelength for HPALDs is assumed to be the same as methacrolein (Peeters et al., 2009, Wennberg et al., 2018, Schwantes et al., 2020) but with a significantly larger quantum yield (QY). Prather et al (2013) recommends a QY of 0.003 for methacrolein and Liu et al (2017) a QY of 0.55 for HPALDs (both used by Wennberg et al., 2018). To implement the photolysis of these new species, the photolysis frequencies of HPUCARB12 was taken to be the photolysis frequency for methacrolein scaled by the ratio of the QY of HPALDs to the QY of methacrolein, the same approach used by Schwantes et al (2020) for δ-HPALDs. A scaling of 0.5 was applied to the photolysis frequency of HUCARB9 in agreement with the MCM v3.3.1.

In addition to the updates to isoprene chemistry, CRIv2.2 has had the rate coefficients for many organic and inorganic reactions updated to bring the mechanism into agreement with the MCM v3.3.1 and IUPAC. These affect the overall chemistry in three major ways. The first involves the major reactions of the excited oxygen radical, $O(^1D)$. The rate constants of $O(^1D)$ with $H_2O$, $O_2$ and $N_2$ changed by -3%, -1% and +20% respectively to bring them into agreement with the current IUPAC values (http://iupac.pole-ether.fr). This also means the rate constant of $O(^1D)$ with $N_2$ became much closer (within ±1.5%) to that used in ST (Archibald et al., 2020a) and rate constants for the reactions with $O_2$ and $H_2O$ also move closer to those used by ST. The result of this is a reduction in the fraction of $O(^1D)$ reacting with $H_2O$ by 10-15%, thus lowering OH production while also reducing $O_x$ loss via this pathway.

The second involves multiple inorganic reactions of nitrated species. The formation rate constants for PAN-type species (species with peroxyacyl nitrate functionality), $HONO_2$, $HO_2NO_2$ and $N_2O_5$ changed by around -45%, -15%, -45% and +50-75% in the troposphere respectively. The change for PAN brought its formation rate constant much closer to that used in ST (within ±7%) and this was also the case for $HONO_2$ and $HO_2NO_2$ formation. The rate constant of $HO_2$ + NO, the single biggest production source of $O_x$, decreased by 4%.

Finally, the rate constants for most $RO_2$ + NO and $RO_2$ + $NO_3$ reactions have been changed by +12.5% and -8%, respectively while maintaining the same temperature dependence. This is likely to have a smaller impact that the other chemistry changes but, at the margins, will make reactions with NO more competitive with the isomerisation reactions of the ISOPOO.
The implementation of CRI v2.2 by Khan et al (2021) in STOCHEM model, while including the updates to isoprene chemistry and the $RO_2$ + NO and $RO_2$ + $NO_3$ reactions, did not feature updates to the rate constants for $O(^1D)$ with $H_2O$, $O_2$ and $N_2$ or the inorganic nitrogen reactions. Therefore, even in low altitude terrestrial conditions where isoprene $HO_x$-recycling tends to dominate the change in OH, comparison between Khan et al (2021) and the results of this work must be caveated with the changes to the inorganic chemistry.

In addition to the chemistry changes, updates are made to the photolysis of several species. Two additional photolysis reactions of glyoxal (CARB3 in the CRI mechanisms) were added as well as updates to the photolysis parameters for HCHO and EtCHO (propionaldehyde). The wavelength bins of the product of cross-section and quantum yield used by FAST-JX (Telford et al., 2013) used were updated to the v7.3 values from Prather et al (2015) for HCHO and $C_2H_5CHO$. The photolysis of CARB3, which had previously been estimated in CS by a scaling of HCHO photolysis (Archer-Nicholls et al., 2020), is replaced with the glyoxal photolysis for 999 hPa from v7.3 of Prather et al (2015). This reaction does exhibit a modest pressure dependence but one which has not been incorporated into FAST-JX at the current time.

In addition to the changes to the chemistry and photolysis, updates to the wet deposition scheme were implemented to both CS and CS2 schemes. The previous approach of applying parameters for standard surrogate for other species with the same functional groups (e.g. EtOOH was used for most hydroperoxides), as described in Archer-Nicholls et al (2020), was updated to use either data for the precise species (taken from Schwantes et al., 2020) or a more closely related-surrogate. The changes to the wet deposition parameters are detailed in Table S1 of the supplement and, as they were applied to both CS and CS2 mechanisms, they are unlikely to have a significant influence on the inter-mechanism difference. No changes were made to the dry deposition scheme in this work.

## 3. Model Runs

All model runs were performed using the United Kingdom Chemistry and Aerosols Model (UKCA) run at a horizontal resolution of 1.25° × 1.875° with 85 vertical levels up to 85 km (Walters et al., 2019) and the GLOMAP-mode aerosol scheme which simulates sulfate, sea salt, BC, organic matter, and dust but not currently nitrate aerosol (Mulcahy et al., 2020). In this setup, the inert chemical tracer Sec_Org, which condenses irreversibly onto existing aerosol, was produced at a 26% yield solely from reactions of α-pinene and β-pinene with $O_3$, OH and $NO_3$ with the enhanced yield applied to account for a lack of SOA formation from isoprene or anthropogenic species (Mulcahy et al., 2020).

The runs in this work fell into two distinct categories. Firstly, short runs (generally 1-2 months, Table 3) with higher frequency (hourly) output using the ST, CS and CS2 chemical mechanisms were performed to evaluate each mechanism's performance against the observational data. Secondly, longer runs (2-5 years, Table 4) with monthly output using the CS and CS2 chemical mechanisms (or variants of CS2 for sensitivity tests) were conducted to facilitate a rigorous comparison of the global chemical composition (Table 4).

Temperature and horizontal wind fields were nudged (Telford et al., 2008) in all model runs to atmospheric reanalyses from ECMWF (Dee et al., 2011) to constrain the simulations to consistent meteorology, thus preventing diverging meteorology adding to the differences resulting from the chemical mechanisms and replicating as closely as possible the atmospheric conditions experienced when the observations were recorded. Nudging only occurred above ~1200 m in altitude and thus the majority of the planetary boundary layer was not nudged. The model runs were atmosphere-only with prescribed sea surface temperatures (SSTs). $CO_2$ is not emitted but set to a constant field while methane, CFCs and $N_2O$ are prescribed with constant lower boundary conditions, all at 2014 levels (Archibald et al., 2020a).

The emissions used in this study the same as those from Archer-Nicholls et al (2020) and are those developed for the Coupled-Model Intercomparison Project 6 (CMIP6) (Collins et al., 2017). Anthropogenic and biomass burning emissions data for CMIP6 are from the Community Emissions Data System (CEDS), as described by Hoesly et al. (2018). For the short runs, timeseries anthropogenic and biomass burning emissions were used for all ST runs and all CRI runs up to 2015. For the runs done for comparison to observational date recorded at the Z2F site new Manaus in 2016 (see Tables 3, 5) , timeslice 2014 emissions were used due to a lack of post-2015 CRI emissions although the impact of the difference is expected to be minimal.

All longer runs used time slice 2014 emissions for anthropogenic and biomass burning emissions. Oceanic emissions were from the POET 1990 dataset (Olivier et al., 2003) and all biogenic emissions except isoprene and monoterpenes (see Section 3.3) were based on 2001-2010 climatologies from Model of Emissions of Gases and Aerosols from Nature under the Monitoring Atmospheric Composition and Climate project (MEGAN-MACC) (MEGAN) version 2.1 (Guenther et al., 2012) and are discussed further in Section 3.3. A full description of the emission sources for each emitted species is given in Table S2.

All mechanisms used the same raw emissions data. However, the additional emitted species required by CS and CS2 means the total mass of emitted organic compounds is greater in CS and CS2 and the lumping of species for emissions is also different. The approach and consequences are discussed in Archer-Nicholls et al (2020).

### 3.1 Short runs for model-observation comparisons

The runs performed for comparison to observations are detailed in Table 3 and correspond to an observational dataset described in Section 4 and Table 5. All runs were spun-up for a minimum of three months. For most of the runs, hourly model output was used so as to allow for detailed comparison with observations. The only exception were the runs performed for the comparison to the Isoprene Column data ("Isoprene Column" Table 3) where monthly means were used.

### 3.2 Longer runs for mechanistic intercomparison

The longer runs (Table 4) were designed with the primary aim of examining the consequences of the mechanism changes between CS and CS2 and followed an approach similar to that used by Archer-Nicholls et al (2020). These

runs also served a secondary purpose as they enabled longer term comparison to observations for several species. We ran two 5 years nudged runs (1 year spin up, 4 years analysis) with the CS and CS2 mechanisms. In addition, five 2-year sensitivity runs (1 year spin up, 1 year analysis) were performed to analyse the impact of the individual changes to the isoprene scheme, the $O(^1D)$ reactions, inorganic nitrogen reactions, the $RO_2$+NO/$NO_3$ reactions and the photolysis reactions as discussed in Section 2. These sensitivity tests featured mechanisms based on the CS2 mechanism but each had a different feature which was reverted to that found in CS.

CS2_O1D used the old rate constants from CS for the reaction of $O(^1D)$ with $N_2$, $O_2$ and $H_2O$. CS2_inorgN used the rate constants from CS for the formation of $HONO_2$, $HO_2NO_2$, PANs, HONO and $N_2O_5$ as well as for the reactions of $HO_2$ + NO, OH+$MeONO_2$, OH+ PAN and OH+MPAN.

CS2_isoprene followed as closely as possible the isoprene reactions from CS with the major change being the omission of the isomerisation reactions of RU14O2 and subsequent production of HPALDs and other species which are key for $HO_x$ recycling.

In CS2_RO2_N, the rate constants for the $RO_2$ + NO and $RO_2$ + $NO_3$ reactions were reverted to those used in CS which led to a 12.5% decrease and 8% increase, respectively for the vast majority of these reactions. Where branching ratios changed between CS and CS2, the CS2 branching ratios were maintained and the rate constants scaled accordingly.

Finally, CS2_photo used the parameters and reactions from CS for the photolysis of CARB3 (glyoxal), HCHO and EtCHO and was performed to evaluate the impact of update to photolysis (see SI Section S6).

Each sensitivity test, when compared to the CS2 run, provides information as to the impact of the change of the respective section of the mechanism (when taken in isolation); for example, the impact of the changes to the rate coefficients of $O(^1D)$'s reactions is examined by comparing the CS2 and CS2_O1D runs.

A full description of the changes to reactions and rate constants for each sensitivity test is given in the supplement Section S2. The changes to the photolysis were found to have a minimal effect on atmospheric composition compared with the other sensitivity tests and is described entirely in the supplement. The analysis of the longer runs is discussed in Section 5.

### 3.3 Biogenic Emissions

This work used the interactive Biogenic Volatile Organic Compound (iBVOC) emissions system (Pacifico et al,. 2012) for isoprene and monoterpenes, the standard approach for UKESM's contributions to CMIP6 (Sellar et al., 2019). Emissions of isoprene and monoterpenes are calculated interactively based on temperature, photosynthetically active radiation (PAR) and plant functional type for each grid cell. While a diel cycle for isoprene is standard in UKESM,

iBVOC has the advantage of also simulating a diel cycle of emissions for monoterpenes, leading to improved model performance relative to observation (see Section 4). The dependence on temperature and PAR means that emissions of BVOCs differ slightly between runs and thus between mechanisms. However, nudging inhibits considerably divergence of surface temperature between comparative runs and so the differences between emissions were <5% and typically 1-2%, significantly smaller than the differences caused by the mechanisms.
Monoterpenes emissions were speciated in a 2:1 α-pinene : β-pinene ratio as used in Archer-Nicholls et al (2020).

There are temporal and spatial disparities between using iBVOC emissions and offline emissions, such as the MEGAN-MACC dataset ((Sindelarova et al., (2014), as used by Archer-Nicholls et al., (2020)), which could affect conclusions about mechanism-observational biases. These differences are discussed in more detail in SI Section S. In short, for the ZF2 Brazil, ATTO and Borneo sites for the periods considered, the isoprene and MT emissions were higher when using the iBVOC approach than for MEGAN-MACC (Figs. S1, S2).

**4. Comparison with Observations**

The shorter UKCA models runs listed in Table 3 were used to evaluate mechanism performance against 6 high frequency observational datasets (3 surface/near-surface and 3 aircraft campaigns) from the Amazon, Borneo and the South East USA, all important regions for BVOC production. In addition, satellite-derived isoprene columns (Wells et al., 2020) were compared to model output (Isoprene Column, Table 3). Monthly mean data from the longer CS and CS2 runs (Table 4) for $O_3$, CO and $HONO_2$ were also compared to a range of observational data. A summary of the observation datasets is given in Table 5 and locations of the surface and airborne campaigns shown in Fig S3.

Diel profiles for multiple species were calculated from the three surface/near-surface sites and the vertical profiles were calculated from the ATTO site.

The three flight campaigns considered were the October 2005 Amazon GABRIEL campaign (Butler et al., 2008), the July 2008 Borneo Facility for Airborne Atmospheric Measurements (FAAM) (Hewitt et al., 2010) and the Studies of Emissions and Atmospheric Composition, Clouds and Climate Coupling by Regional Surveys ($SEAC^4RS$) flight campaign over the South East USA in August - September 2013 (Toon et al., 2016). Hourly model output corresponding to the days and times of the flights was used for the mechanism-observation comparison for each campaign. Model and observational data were binned into 250m/500m altitude bins and median values for the variables of interest across the whole region for a given altitude bin were considered. For the $SEAC^4RS$ comparison, observational data were also filtered to exclude urban plumes ($NO_2$>4 ppb), fire plumes (acetonitrile>0.2 ppb) and stratospheric air ($O_3$/CO > 1.25) while missing data or data flagged as exceeding the limit of detection were not used and data flagged as a lower limit of detection were set to zero as done in Schwantes et al (2020). Estimated limits of detection are shown for relevant species for the GABRIEL and FAAM campaigns.

The performance of each mechanism is now described for the key species e.g. $O_3$, $HO_x$, isoprene, certain isoprene oxidation products and monoterpenes. A brief commentary about other species including $HONO_2$, CO, PAN, HCHO, MeCHO, EtCHO and acetone is given in the supplement.

**4.1 Ozone**

CS2 exhibits a modest increase in $O_3$ (~1-2 ppb) over CS at all surface sites (Fig. 1), exacerbating the existing high surface bias of CS, whose drivers were discussed in Archer-Nicholls et al (2020), and the smaller high bias of ST. On a diel basis, the mechanisms are able to replicate the shape of the diel cycle at the ZF2 site (with similar diel profiles at the ATTO site) but perform less well in Borneo, simulating pronounced diel cycles with a high bias compared to much more muted cycles from observation.

An increase of ~1-4 ppb relative to CS is also exhibited by CS2 for monthly mean $O_3$ when both mechanisms are compared to observational data at 10 locations from pole to pole at 4 pressure levels (250, 500, 750 and 900 hPa) (Fig. S4). CS2 reduces the low bias in polar regions but increases CS's high bias in the tropics and Eastern US.

Model high biases are also observed from flight data comparisons (Figs. 2(b,f), S6(a)). In the Amazon, where the observed and modelled NO vertical profiles agree well (Fig. S6(e)), there is little difference between the three mechanisms. Each exhibits the greatest high bias at low and a smaller high bias in the free troposphere. CS2 exhibits a high bias of 15-20 ppb for the SEAC$^4$RS campaign (Fig. S6(d))), with perhaps some influence from the low altitude $NO_2$ model high bias. In Borneo, all mechanisms exhibit a roughly consistent high bias of ~20 ppb for ST increasing to 30 ppb for CS2. Interestingly, all the mechanisms simulate a significant low bias for $NO_2$ (Fig. S6(f)) which may indicate biomass burning events which are not simulated, something which might be expected to promote higher ozone concentrations.

**4.2 $HO_x$**

Modelled surface OH increases in all locations from ST through CS to CS2 with a significant increase in midday OH from CS to CS2 (Fig. 1). In Borneo, OH is consistently low biased in the three mechanisms but the best comparison is exhibited by CS2 where the mean diel bias compared to ST and CS decreases by 43-50% and 24-40%, respectively over the period considered. The drivers of the $HO_x$ change are explored further in Section 5.

Surface $HO_2$ was also simulated to increase in all locations from ST to CS to CS2. Significant high bias was simulated in Borneo (the only observational dataset) (Fig. S7) for the CRI mechanisms, including at night. The simulated ratio of $HO_2$ to OH is highly biased in all mechanisms. However, it is best simulated in CS2, indicating that the increase in OH is much larger than that for $HO_2$. It should be noted that none of the mechanisms at present include the heterogenous reactions of $HO_2$ and their inclusion, which will be addressed in future work, should reduce the $HO_2$ high bias.
The comparison of modelled $HO_x$ to observation is complicated by large discrepancies in key reaction partners. Furthermore, relative to observed values of 100-130 ppb, CO in ST in Borneo is highly biased by 13 ppb and 27 ppb while CO in the CRI mechanisms exhibits larger biases of ~35-50 ppb and ~50-60 ppb during April-May and June-July, respectively (Fig. S7). These high biases would enhance modelled $HO_2$ at the expense of OH, potentially explaining the modelled low biases in OH. Indeed, the OH model low bias is greater in the June-July period. This highlights the complexity of model-observation comparisons: the CRI mechanisms may well simulate secondary CO production from isoprene more accurately than ST but other model biases, for example in emissions of CO, NO and isoprene, can lead to the CRI mechanisms appearing worse. Nevertheless, if the CO high bias is reduced in future, we might reasonably assume the modelled OH will improve still further.

### 4.3 Isoprene

Modelled isoprene from all three mechanisms was compared to surface observations, flight campaign data and isoprene columns measured by satellite.

#### 4.3.1 Isoprene Surface Measurements

CS2 yields the best model-observation comparison for surface isoprene on a daily basis in all locations (Fig. 1 (k-o)). CS2 reduces the high bias in the diel profiles by 50-60% relative to ST and 20-40% to CS at the Z2F, ATTO and Borneo sites, driven by the elevated OH concentrations

In most locations the model simulates, to a greater or less extent, a "twin peak" isoprene profile with a sharp rise around 7:00 LT and a second, smaller peak at 19:00 LT. This was most pronounced in the Amazon dry season (ATTO Sept 2013). The morning peak is likely to be due to a combination of the sharp rise in simulated isoprene emissions which starts at 6:00-7:00 am LT, outweighing the concurrent rise in OH, and an underestimation in the model of the rate of BL height growth which can trap isoprene close to the surface, causing a buildup. By contrast, observed isoprene concentrations exhibit a much slower morning growth reaching a peak in early afternoon. While this "out-of-phase" behaviour is unlikely to be the sole driver of model-observation difference, it will play a role since isoprene chemistry occurs on the time scale of ~1-2 hours and atmospheric oxidising capacity varies throughout the day.

Over the lowest 80 m at the ATTO site, all mechanisms are high biased in the daytime (9:00-15:00) and nighttime (21:00-3:00) (Fig. S8 (a-d)) with CS2 exhibiting the smallest bias but produce similar isoprene vertical gradients to observations. The effect of boundary layer height was further considered by looking separately at the periods 6:00-8:00 LT and 17:00-19:00 LT (Fig. S8 (e-h)). In contrast to the daytime and nighttime periods, during the 6:00-8:00 period the simulated isoprene gradient is significantly more negative than the observation, indicating less vertical mixing and similar results are seen with the MT profile (Fig. S8 (m-p)). This is most noticeable in September where the largest morning peak is seen in the diel profile for both species and lends support to the theory that the simulated BL height is not increasing as quickly as in reality, leading to more isoprene and MT being trapped at low altitude. Smaller differences between observed and simulated isoprene and MT vertical gradients are seen during 17:00-19:00

LT, coinciding with smaller evening peaks in the diel profiles. This suggests the reduction in BL height is more accurately simulated than the morning increase.

The major drivers of the remaining model-observation difference are likely to be the concentrations of oxidants (despite the increases seen in CS2, OH remains low biased in Borneo) and the emissions of isoprene (including the modelled vs. actual diel cycle). The concentrations of isoprene and other species also vary significantly through and above the tree canopy, as shown by the ATTO measurements (Fig. S8), and the global model resolution is not high enough to resolve the vertical gradient of species in the canopy. When testing the CRI v2.2 in STOCHEM-CRI with isoprene emissions from the MEGAN-MACC inventory, Khan et al (2021) noted that halving the isoprene emissions reduced the model-observation disagreement significantly and attributed the model high bias in their work to high biases in the emissions of isoprene.

**4.3.2 Isoprene Flight Measurements**

Model-observation comparisons of isoprene vertical-profiles extending into the boundary layer and into the free troposphere reveal quite a different story from the surface analysis (Fig 2 (a, e, h)).

Despite being high biased at the surface and at low altitude, simulated isoprene vertical profiles over the Amazon and Borneo rapidly show a low bias as latitude increases. There are likely two reasons for this. The first is the vertical mixing, already discussed in relation to the isoprene and MT surface diel cycles. Secondly, for the Amazon and Borneo campaigns only estimated detection limits (0.1 ppb in both cases) could be used. This has the effect of biasing the median of the observational data to higher values as very low values are ignored. In the SEAC$^4$RS campaign, all data points flagged as below the detection limit were set to zero, mitigating this issue. The enhanced oxidative capacity of CS2 at low altitude results in the lowest simulated vertical concentrations among the three mechanisms but the general low bias above the surface is an issue faced by all mechanisms, suggesting it is not just down to modelling of the chemistry.

**4.3.3 Isoprene Columns**

To consider isoprene on a global scale, monthly modelled isoprene columns for all mechanisms are compared to satellite observations from January, April, July and October 2013 (Wells et al., 2020) (Fig. 3).

Significant variation in model bias is exhibited between the mechanisms with ST exhibiting the highest isoprene columns and CS2 the lowest. In South America CS2 exhibits the smallest bias while the ST columns are over double the observed values for April and July. CS and CS2 exhibit the smallest biases in Africa and Southeast Asia respectively. The low biases in North America (~ $0.7\text{-}1.5\times10^{15}$ molecules cm$^{-2}$), Europe (~ $0.5\text{-}2.7\times10^{15}$ molecules cm$^{-2}$) and Central Asia (~ $0.1\text{-}1.1\times10^{15}$ molecules cm$^{-2}$) are quite consistent across the mechanisms and, in some cases almost equal in magnitude to the observed columns, which suggests the bias is driven more by insufficient emissions rather than the chemistry scheme in these locations.

CS and CS2 yield lower isoprene columns and generally smaller model biases than ST. This comparison highlights the significant influence of the different chemistry schemes on the simulated isoprene column and thus the considerable challenges of determining isoprene emissions via top-down approaches using back-calculation from observed concentrations or column values: different chemistry schemes will lead to different emission estimates.

### 4.4 Isoprene Oxidation Products

During the GABRIEL flight campaign, the well-known isoprene oxidation products MACR and MVK were measured via PTRMS and combined as isoprene oxidation products. These species, along with the ISOPOOH, were also measured at the ATTO tower via PTRMS and are compared with model data.

At the ATTO site, all mechanisms are largely high biased but CS2 produces the best comparison to observations for both diel and vertical profiles (Figs. 1, S9, 11). CS2 also yields the smallest high bias for the ratio of isoprene oxidation products (isop_ox) to isoprene (a metric less sensitive to discrepancies between actual and modelled isoprene emissions) in the Amazon (Figs. 1, S9, 11). Despite the greater oxidising capacity of the PBL in the CS2 simulations, the isop_ox concentrations are lower. This is attributed to the fact that in the relatively low $NO_x$ environment around the ATTO tower, the isomerisation reactions of the isoprene peroxy radical are particularly important and favour the production of HPALDs and other species over MACR, MVK and ISOPOOH.

Relative to the GABRIEL flight data (Fig. 2(d)), the ratio of isop_ox to isoprene is high biased in all mechanisms albeit with the CRI mechanisms exhibiting a smaller bias than ST.

### 4.5 Isoprene Nitrate, IEPOX and HPALDs

The isoprene oxidation products HPALDs and IEPOX, unique to the CS2 mechanism in this study, are compared, along with isoprene, ISOPOOH and the isoprene nitrate (Fig. S6), to observational data from the SEAC$^4$RS campaign over the Southeast USA. Modelled isoprene (Fig 2(h)) exhibits a significant low bias, in line with the isoprene column analysis (Fig. 3) and is attributed to insufficient emissions. Unsurprisingly, ISOPOOH (Fig 2(i)), the isoprene nitrate (Fig. S4(c))) and HPALDs (Fig 2(j)) are also low biased. However, IEPOX (Fig 2(j)) compares favorably to observation.

The apparent good performance of IEPOX, despite the significant low biases of isoprene and its direct precursor ISOPOOH, is likely to be due to a missing sink to the aerosol phase. IEPOX is readily lost to aerosol by reactive uptake (Nguyen et al., 2014, Nguyen et al., 2015, Allan et al., 2014); a process featured in Schwantes et al (2020) (who simulated lower IEPOX concentrations) but not in UKCA. The rate constant for IEPOX's production from ISOPOOH is ~30% lower than that used by a mechanism of similar complexity, MOZART TS2 (Schwantes et al., 2020) while IEPOX's loss via OH has a similar rate constant to MOZART TS2. Including reactive uptake of IEPOX

in future updates may reduce this high bias. The processing of IEPOX is unlikely to affect $HO_x$-recycling as much as HPALDs, however its importance to SOA formation means it will be a focus of future work.

The low bias of HPALDs, also simulated to a lesser extent in Schwantes et al (2020) who used isoprene emissions from the MEGAN v2.1, is important given its role in $HO_x$-recycling via photolysis. There remains uncertainty in HPALD photolysis frequencies. In this work simulated HPALD destruction is dominated by reaction with OH and photolysis which are roughly equal ascending to 2.5 km whereupon OH's importance grows rapidly at the expense of photolysis. To test the impact of photolysis uncertainty on the bias, two further runs were performed with the photolysis frequency of HPALDs scaled by 0.5 and 3, respectively. These tests change HPALD concentrations in the lowest 2 km by +30% and -50% (Fig. 2(k), respectively, suggesting concentration of HPALDs is dependent on the photolysis frequency of HPALDs, which is not currently well constrained.

Interestingly, these scaling tests only change low altitude OH by ~2-3% in the south east USA, suggesting the uncertainty in HPALD photolysis from the current approach may not have a huge impact on oxidants in this region although this may in part be due to the modelled isoprene and HPALD low biases (Fig. 2(h,k)). Furthermore, the fact that the modelled photolysis frequency of methacrolein here is low biased by a factor of 2.5-3 (not shown) suggests that, if further changes to the HPALD frequency are made in future, any potential reductions in methacrolein frequency should be scrutinised carefully. Nevertheless, constraining HPALD photolysis further will be a key focus of future work. A lack of OH measurements prevents attempts to constrain the OH loss pathway.

Evaluating HPALD production is also challenging since observations of ISOPO2 were not measured. Over the relevant temperatures, the rate constant for HPALD production in CS2 is 6-14 time greater than the equivalent used by Schwantes et al (2020) which would, if anything, make a low bias less likely. The sensitivity of HPALD production to the concentrations of the bimolecular reaction partners of ISOPO2 (e.g. NO) can also lead to resolution issues with the model: regions with high and low NO concentration treated as a single region within the model (model grids can be up to ~125 km wide at the equator) with moderate [NO], suppressing HPALD formation (see Schwantes et al., 2020). A commentary on the global distribution of HPALDs and IEPOX is given in Section 5.

### 4.6 Monoterpenes (MT)

Simulated surface diel monoterpene profiles (Fig. 1) are characterised by early morning and evening peaks which are not present in observations. As discussed in relation to the isoprene diel cycle, the morning peak is probably caused by a combination of the simulated emissions increasing too early and a delayed evolution of the simulated BL height, trapping large quantities of monoterpenes close to the surface (Fig. S8). The evening peak coincides with a reduction of simulated OH to near zero and therefore is probably driven by oxidant reduction as well as a reduction in the BL height. Around midday the mechanisms do a better job in most locations with the lower values in the CRI mechanisms driven by the greater oxidant concentrations. In 4 of the 5 locations, CS2 yields the smallest model bias although is it acknowledged that other issues, such as the BL dynamics, need attention.

## 5. Comparison to CRI-STRAT

The performance of the CS mechanism to the simpler ST mechanism was discussed in detail in Archer-Nicholls et al (2020). Here we describe chemical composition of the atmosphere simulated by CS2 relative to that from CS using the longer model runs summarised in Table 2. Particular attention is paid to $O_3$ and its production and loss fluxes, $HO_x$, isoprene and monoterpenes, the isoprene oxidation productions IEPOX and HPALDs, nitrated species ($NO_y$) and the potential impacts to aerosols. Changes to CO and HCHO are discussed in the SI Section S5.

### 5.1 $O_x$

As in Archer-Nicholls et al (2020), the change to $O_3$ was analysed by considering the sum of odd oxygen, $NO_2$ and its reservoir species, termed $O_x$, and defined in Eq. (1).

$$O_x = O + O_3 + NO_2 + 2N_2O_5 + 3NO_3 + HONO_2 + HO_2NO_2 + PANs \quad (1)$$

Tropospheric $O_3$ burden increases by 8% from 328 Tg in CS to 354 Tg in CS2. Much of the free troposphere exhibits increases of 2-6 ppb (~6-14%) in $O_3$ with large parts of the tropical troposphere increasing by more than 4 ppb (Fig. 1). This increase is driven chiefly by a 1.3% decrease in $O_x$ chemical destruction, resulting in an 12% increase in net chemical $O_x$ production. The sensitivity tests (Table S3) reveal the update to the isoprene mechanism only has a minor effect on $O_3$ burden (~2 Tg decrease) while the changes to $O(^1D)$ and inorganic nitrogen reactions each yield increases of 17 Tg (when considered in isolation) with greater impacts in the lower and upper troposphere, respectively (Fig. S16). The changes to the $O_3$ burden in the sensitivity tests do not sum to the total 26 Tg increase from CS to CS2 which indicates a degree of interplay between the different updates, an unsurprising result given ozone's central role in tropospheric chemistry.

$O_x$ lifetime, defined as the ratio of $O_x$ burden ($B_{Ox}$) to the sum of chemical ($L_{Ox}$) and physical ($D_{Ox}$) $O_x$ loss fluxes (Eq. 2) (Young et al., 2018, Archibald et al., 2020b), increase by 8% equivalent to 18.8 days in CS2, while ozone production efficiency (OPE), defined as moles of $O_x$ produced ($P_{Ox}$) per mole of $NO_x$ emitted ($E_{NO}$) (Eq.3) (Archer-Nicholls et al., 2020) increases negligibly from 33.74 to 33.78.

$$\tau_{O_x} = \frac{B_{O_x}}{L_{O_x} + D_{O_x}} \quad (2)$$

$$OPE = \frac{P_{O_x}}{E_{NO}} \quad (3)$$

$O_3$ below 500 m increases across almost the entire globe with increases of 2-4 ppb (~5-7.5%) over much of Europe, Africa and the Americas and 4-5 ppb over India and China (Fig. 4), exacerbating the existing high bias in CS (Archer-

Nicholls et al., 2020). The sensitivity tests allow this change to be partially decomposed into the different drivers (Fig. S13). The update to isoprene chemistry produces localised increases in $O_3$ over the tropical forested regions of South America, Africa and East Asia of 2-4 ppb: the increase in $O_x$ production via $HO_2$ + NO and $MeO_2$ + NO outweigh the reduction in the non-methyl peroxy radicals ($RO_2$ + NO) pathway (discussed later). While comparison to Khan et al (2021) is difficult given the multiple mechanistic differences, $O_x$ production from $RO_2$ + NO also decreased in their study. The changes to $O(^1D)$ also yield an increase in >1 ppb across the entire globe (due to reduced $O_x$ loss via $O(^1D)$ + $H_2O$) with a larger increase (2-3 ppb) encompassing ~20S-40N. The change to inorganic nitrogen also leads to terrestrial increases of 2-4 ppb from increased $O_x$ production via $HO_2$ + NO and $RO_2$ + NO.

#### 5.1.1. $O_x$ Budget

$O_x$ production and loss fluxes for CS and CS2 are given in Table 6 and the breakdown for the sensitivity tests is given in Table S3. $O_x$ production decreases in CS2 in much of the tropical and SH BL and lower free troposphere but increases in the NH midlatitude BL and tropical high troposphere while $O_x$ loss decreases strongly in the tropical BL and lower free troposphere (Fig. 5). Despite the modest changes to total $O_x$ production and loss fluxes, the story is more complicated than it first appears due to offsetting changes to the key chemical production and loss fluxes.

#### 5.1.2 $O_x$ production

The $HO_2$ + NO pathway represents the largest absolute increase of $O_x$ production (3.2%, Table 6) with particular increases in the NH tropics and mid latitude boundary layer and tropical upper troposphere (Fig. S12). The drivers of this change are complex: the low altitude increases are driven by the significant increases in $HO_2$ (Fig. 6), which exceed 5% in places, while at higher altitude the increase is attributed to a localised 15-20% rise in NO. The sensitivity tests suggest the change to the isoprene scheme (CS2_isoprene) is a key driver in the rise of low altitude $HO_2$ (and thus the flux) while the change to the inorganic nitrogen reactions (CS2_inorgN) also contribute to the increased flux at low altitudes and are chiefly responsible for the increase at higher altitudes.

However, the increase in $HO_2$ + NO is offset by a decrease in the NO + $RO_2$ flux (15.4%, Table 6) where $RO_2$ comprises all peroxy radicals except the methyl peroxy radical, $MeO_2$. This reduction is strongest in the tropical BL and low free troposphere and driven by a significant decrease in the $RO_2$ burden (32%). This burden reduction arises from the isomerisation pathways which inhibit the conversion of the isoprene-derived peroxy radical, RU14O2, to the other peroxy radicals RU12O2 and RU10O2 (via reactions with standard partners such as NO and $NO_3$) by providing competing routes which yield other species whose degradation pathways do not produce further $RO_2$ (Khan et al., 2021). For example, the HPALDs produced are photolysed to hydroxy acetone and unsaturated hydroxy carbonyls which further degrade producing mostly closed-shell products and $HO_2$. This rapid reaction pathway for RU14O2 sees its burden decrease by 35% in CS2 compared to CS and tropical low altitude mixing ratios decline by over 30%. Similar declines in the $RO_2$ + NO flux (15%) and $RO_2$ burden (33%) are seen for CS2 relative to the CS2_isoprene sensitivity test, providing strong evidence that the change to isoprene is driving the change in $RO_2$. Khan et al (2021)

also simulated a reduction in $RO_2$ burden (and a corresponding drop in $O_3$ production via this pathway) although their decrease of 6.5% is less than half the equivalent value (including $MeO_2$) of 15% in this work, likely due to the other differences between the mechanisms used in their work and this study (see Section 1).

The fluxes of NO with $HO_2$, $MeO_2$ and $RO_2$ account for over 99.5% of total $O_x$ production in both mechanisms and the changes in other pathways are an order of magnitude smaller in absolute terms. The reduction in the rate constant for OH + $MeONO_2$ (Section 4.1) reduces $O_x$ production from organic nitrate oxidation significantly while also driving the increase in $O_x$ production from organic nitrate photolysis. The addition of the photolysis of isoprene hydroxy nitrate and the other nitrates RU12NO3 and RU10NO3 make smaller contributions.

#### 5.1.3 $O_x$ loss

The change in $O_x$ chemical destruction is dominated by the reduction in $O(^1D) + H_2O$ reaction (7.2%) which accounts for 54% of $O_x$ loss in CS but only 49% in CS2. In the sensitivity run CS2_O1D, which uses the same $O(^1D)$ rate constants as CS, the $O(^1D)+H_2O$ flux accounts for 54% of $O_x$ chemical loss. As this reaction involves water, the change is strongest in the tropical BL and low free troposphere (Fig. S14).

The increase in $O_x$ loss via $HO_2 + O_3$ (9.1%, Table 6) is driven predominantly by changes to the inorganic nitrogen and $O(^1D)$ reactions while the isoprene scheme is simulated to have little impact. $O_x$ loss via OH + $O_3$ also increases (7.6%) despite the decrease in free troposphere $HO_x$ with the new isoprene chemistry and revised inorganic nitrogen reactions simulated to play important roles. $O_x$ destruction from $O_3$ + alkene reactions decline significantly (39%) yet increase at very low altitudes (<500 m) before decreasing at higher altitudes. This altitude dependence may arise from the enhanced $O_3$ low altitude driving a greater $O_3$ + alkene flux but, at higher altitudes, the depletion of the VOCs by $O_3$ and the elevated OH, means $O_3$ destruction is lower.

### 5.2 $HO_x$

The change to OH shows significant spatial and altitudinal variation, increasing at low altitude over land but decreasing over the oceans and in much of the free troposphere. This stems from the different drivers of OH concentrations and their relative importance in different regions.

At low altitude, the terrestrial increases in OH (Fig. 6(c,d)) are revealed by the sensitivity tests to be driven predominantly by the isoprene scheme: a clear illustration of impact of the $HO_x$-recycling chemistry (Fig. S15). The inorganic nitrogen changes make a smaller contribution to the low latitude OH increase while the $O(^1D)$ changes reduce low altitude OH but this effect is only noticeable over the oceans.

This significant increase in low terrestrial altitude OH is of particular interest in the context of BVOCs and their impact on the chemical composition of the atmosphere. Concentrations in the lowest 500 m increase by 2-3 × $10^5$ $cm^{-3}$ (30-50%) in much of the Amazon with similar changes seen in other tropical regions and the Southeast USA; regions with the greatest emissions of isoprene and BVOCs. The boreal forest regions in North America and Eurasia exhibit modest

increases of up to 10% in places since isoprene emissions are lower (Fig. S15). The influence of the updated isoprene chemistry is further apparent when the $HO_x$ production flux from two of the key new $HO_x$-recycling pathways - photolysis of the HPALD and hydroxy unsaturated carbonyl (HUCARB9) species - is compared to that from $O(^1D)$ + $H_2O$ (Fig. 6f). Over the Amazon and other tropical regions, $HO_x$ flux from this pathway amounts to 20-40% of that from $O(^1D)$ + $H_2O$. The difference in BVOC-driven depletion of oxidant concentrations at low altitudes will be even more pronounced when CS2 is compared with ST which exhibited even lower tropical low altitude $HO_x$ (e.g. Fig. 9, Archer-Nicholls et al., 2020).

However, in much of the free troposphere, OH decreases by 2-10% relative to CS due to the changes made to the $O(^1D)$ rate constants (Fig. S20) which reduces the fraction of $O(^1D)$ reacting with $H_2O$ (Fig. 6(a)). This general decline is reversed in the upper tropical troposphere (10-15 km) where OH increases by up to 15%, driven by an increase in NO stemming from the update to inorganic nitrogen reactions and a smaller contribution from the updated isoprene chemistry (Fig. S16). These free troposphere changes partially reverse the changes simulated between CS and ST (Fig. 6, Archer-Nicholls et al., 2020). In that comparison, tropical free troposphere OH (~2-6 km) increased in CS relative to ST by $0.5\text{-}2\times10^5$ $cm^{-3}$ while here CS2 yields a decrease in the same location of $0.25\text{-}1\times10^5$ $cm^{-3}$ compared to CS. In the upper tropical troposphere, CS decreased OH by $1\text{-}4\times10^5$ $cm^{-3}$ relative to ST while CS2 exhibits an increase of $0.25\text{-}1.5\times10^5$ $cm^{-3}$ in the same region compared to CS. Thus, the distribution of free troposphere OH in CS2 is more similar to that in ST than the CS distribution is.

Overall, the reduction in the free troposphere OH outweighs the increases elsewhere with the tropospheric air mass-weighted concentration and burden of OH decreasing in CS2 by 1.5% and 0.49% respectively. This is in sharp contrast to the 6.4% increase in burden simulated by Khan et al (2021). However, in the CS2_O1D sensitivity test the OH burden increases by 6.6 % relative to CS allowing us to be confident that this discrepancy between Khan et al (2021) and this work is down predominantly to the differing $O(^1D)$ rate constants. Despite the increase in surface OH, the net reduction in tropospheric OH yields a 2.3% increase in methane lifetime from 7.43 to 7.60 years (Table 7), also in contrast to the 0.5 years decrease in methane lifetime simulated by Khan et al (2021). However, the isolated change to isoprene chemistry, given by the comparison of CS2 and CS2_isoprene, causes a methane lifetime decreases of 2.2% due to the enhanced low altitude OH.

$HO_2$ also increases at low altitude (up to 6-8% at the surface, Fig. 6(b)), driven primarily by the new isoprene chemistry, yet this increase extends much further into the free troposphere than OH, reaching nearly 5 km above the equator. $HO_2$ decreases in the rest of the free troposphere, partially from $O(^1D)$ changes, and does not exhibit the high increase shown by OH, rather declining by 6-8% in the tropical high troposphere resulting in a burden decrease of 0.7%. The greater increase in low altitude $HO_2$ (than for OH) is likely to be due in part to co-located increases in CO of 3-6 ppb (see SI and Fig. S21(a)).

**5.3 BVOCs**

The interactive nature of iBVOC emissions led to average isoprene emissions being 0.36 Tg yr$^{-1}$ (0.06%) lower in CS2 while monoterpene emissions were 0.05 Tg yr$^{-1}$ (0.05%) lower. However, these differences are dwarfed by the reductions in the burdens of isoprene, $\alpha$-pinene and $\beta$ -pinene of 26%, 18% and 15%, respectively.

Isoprene mixing ratios averaged over the lowest ~ 100m decrease by 1-3 ppb (~10-30%) in large parts of South America, Africa and South East Asia (Fig. 7). The greater terrestrial low altitude $HO_x$ increases the OH-initiated oxidative flux of isoprene by 3.5 %, attributable almost entirely to the updated isoprene scheme. However, this is actually outweighed by a 23% decrease in isoprene destruction by $O_3$ while oxidation via $NO_3$ increased by 3.7%. Despite the modest global increase, isoprene oxidation is confined even more to low altitude regions (Fig. 7), a feature also simulated by Karset et al (2018) (Fig. 8). This also results in lower mixing ratios throughout the whole troposphere (Fig. 7).

$\alpha$-pinene's chemical destruction by OH, $O_3$ and $NO_3$ changed by 7.5%, -6.3% and -0.8% respectively leading to a total flux increase of 0.05 Tg yr$^{-1}$ (+0.05%). The corresponding changes for $\beta$-pinene with OH, $O_3$ and $NO_3$ were 7.6%, 5.8% and 12.9% with a total increase of 3.59 Tg yr$^{-1}$ (+7.9%).

The reductions to these BVOC burdens are greater than those simulated by Khan et al (2021) of 17%, 4% and 9% for isoprene, $\alpha$-pinene's and $\beta$-pinene respectively. However, Khan et al (2021) simulated a reduction in $O_3$ over tropical regions and a much smaller increase in $NO_3$ burden (1%) which would have resulted in significantly lower BVOC destruction fluxes, particularly for the monoterpenes. As discussed in Section 4, CS2 simulates a reduction in the model high bias of surface isoprene and, to a lesser extent, monoterpenes, compared to CS and ST.

**5.4 HPALDs and IEPOX**

While a comparison cannot be made between CS and CS2 for HPALDs and IEPOX, their importance for $HO_x$-recycling and SOA formation respectively means examining their global distribution is still useful. Both species follow the surface distribution of isoprene closely (Fig. 8) with IEPOX concentrations typically an order of magnitude greater than HPALDs, something also reflected in their burdens (0.39 Tg and 0.02 Tg, respectively). As discussed in Section 4, loss of IEPOX to aerosol via reactive uptake is not currently modelled and simulated concentrations will decrease once this process is included. Indeed, accurate modelling of IEPOX and its contribution to SOA has been suggested to be important in future climate scenarios (Jo et al., 2021) which highlights the benefits of including IEPOX in CS2 but also the need for careful consideration of how aerosol uptake is modelled. Simulated advection up to the upper tropical troposphere is clearly seen in the DJF zonal means with potentially important consequences for IEPOX-derived SOA which has been observed in the lower troposphere in flight campaigns (e.g. Allan et al., 2014).

**5.5 $NO_y$**

The distribution of nitrated products ($NO_y$) between reactive ($NO_x$) and reservoir species ($NO_z$) changes between CS and CS2 and is detailed in Table 8. Here we use the standard definitions of $NO_x$, $NO_z$ and $NO_y$ (Archer-Nicholls et al., 2020) (Eq. 4,5,6):

$$NO_x = NO + NO_2 \quad (4)$$

$$NO_z = NO_3 + 2N_2O_5 + HONO_2 + HO_2NO_2 + ClONO_2 + BrONO_2 + PANs + RONO_2 + CH_3O_2NO_2 + Nitrophenols \quad (5)$$

$$NO_y = NO_x + NO_z \quad (6)$$

($RONO_2$ comprises alkyl nitrates, hydroxy nitrates and hydroperoxy nitrates while PANs comprises all species with the peroxy acetyl nitrate functionality).

The $NO_y$ burden decreases by 4.8% (in terms of mass of N), driven primarily by a 20% decline in PANs. However, the $NO_x$ burden increases by 4% with the widespread increase in the tropical high troposphere of 10-20 ppt (up to 25%) outweighing the reduction in the NH midlatitude PBL (10-50 ppt, 1-2.5%) (Fig. 9(a)). The increase in $NO_x$ and the reduction in $NO_y$ leads to the fraction of $NO_y$ as reactive Nitrogen increasing by 9% and the associated increases to the $O_3$ production, particularly in the free troposphere, are identified in Section 5.1. The sensitivity tests revealed the high-altitude $NO_x$ rise to be driven predominantly by the change to the inorganic nitrogen with a smaller contribution from the updated isoprene scheme (Fig. S17).

The 6% reduction in $NO_z$ burden is dominated by the decrease in PANs which exceeds 40 ppt in most of the 40N-40S troposphere (Fig. 10(e)). The decrease in the PANs formation rate constant discussed in Section 2 is not the principal driver of this reduction despite reducing by 40% in much of the troposphere. For the single year used for the sensitivity tests, the PANs burden in CS2_inorgN (featuring the larger formation rate constant) (0.292 TgN) is much closer to that in CS2 (0.290 TgN) than in CS (0.364 TgN). A more important factor is the reduction in the PAN-precursor acyl peroxy radical ($MeCO_3$), driven by the updated isoprene chemistry, whose burden decreases by over 20% in both CS2 and the sensitivity test CS2_inorgN. This dependency is clearly illustrated by the fact that the isolated change to the inorganic nitrogen reactions (CS2_inorgN) only produces a small decrease to low altitude PANs while the change to isoprene scheme (CS2_isoprene) yields a much larger decrease in PANs in spatial agreement with the CS2-CS difference (Fig. S18). The PANs burden of 0.317 TgN in the CS2_isoprene test is also closer to that in CS. However, the change in PANs between CS2 and CS is still larger than that simulated from the isolated isoprene chemistry change alone which suggests there are some synergistic effects occurring.

The 0.4% increase in $HONO_2$, including increases of up to 30 ppt in the tropical mid troposphere (Fig. 9(d)), is driven more by the update to the isoprene scheme than the change to inorganic nitrogen reactions (Fig. S19).

The 59% increase in $RONO_2$ burden in CS2 is predominantly due to the significant reduction in the rate constant for the OH-initiated destruction of $MeONO_2$, the principal organonitrate, which brings CS2 into agreement with STRAT-TROP and the most recent IUPAC value. At 290 K, the rate constant is 18 times lower in CS2 and at 250 K 50 times lower, yielding a 3-fold MeONO2 burden increase. The update to the isoprene scheme, when isolated, actually reduces RONO2, despite the introduction of the two new organic nitrates (RU12NO3 and RU10NO3). As discussed in the context of the RO2 burden, this is driven by the added competition from the RU14O2 isomerisation reactions: the flux of the RU14O2 + NO reaction is 15% lower in CS2 than CS. The increase in $RONO_2$ is simulated to be 10-20 ppt in the tropical lower altitude and 2-10 ppt for the rest of the troposphere (Fig. 9(f), S20).

With the significant drop in PANs as a fraction of $NO_y$ (34% to 28%) and the increase in $HONO_2$ and $NO_x$, the breakdown of $NO_y$ in CS2 is closer to that in ST (Archer-Nicholls et al., 2020). The increase in $RONO_2$ is the only major exception to this since ST, which only has two organonitrate species (isoprene nitrate and $MeONO_2$), has a lower $RONO_2$ burden than CS.

### 5.6 Impacts on Aerosols

A key area of future research with the CRI mechanisms will be on their influence on aerosols. The spatial changes to oxidants are also likely to influence secondary organic aerosol (SOA) formation, as discussed in Section 1. In UKCA, SOA is produced from the tracer Sec_Org, a surrogate for the oxidised products of $\alpha$-pinene and $\beta$-pinene which adds to existing organic aerosol with an optional boundary layer nucleation scheme involving Sec_Org and $H_2SO_4$ based on Metzger et al (2010) also available. The Sec_Org burden decreases by 7% in CS2 with noticeable annual variation (DJF -10%, JJA -4%). Despite the burden decrease, within the lowest 500m in the tropics Sec_Org mass concentration increases by 2-10%, driven by an increase its production from $\alpha$-pinene and $\beta$-pinene (Fig. 10(a,b)). Above this region, Sec_Org production and mass concentration decrease and so it appears the greater low altitude oxidative capacity in CS2 leads to greater production of Sec_Org within the boundary layer but lower concentrations above it. This is likely to have an impact on SOA distribution (and lifetime) since deposition and loss to the aerosol phase is greater in the boundary layer due to the steep decline in aerosol surface aerosol density with altitude. Further detailed analysis involving the fluxes of Sec_Org to aerosol and the resulting changes to size and number distributions are beyond the scope of this work but examining wider consequences for SOA, in the context of the BVOC-mediated feedback between the biosphere and climate, will form a key area of future research. It is also worth noting an even more pronounced perturbation to SOA may be seen if isoprene is allowed to produce Sec_Org which is a more realistic approach to simulating SOA (e.g. Scott et al., 2015) and will be explored in future work.

The global perturbation to the oxidation pathways of $SO_2$, another important aerosol precursor, are more modest. From CS to CS2, the oxidative fluxes of $SO_2$ with OH, $H_2O_2$ and $O_3$ change by +0.9%, +0.02% and 1.7%, respectively while the tropospheric sulphate aerosol burden decreases by just 2.3%. However, as with isoprene oxidation and Sec_Org production, the burden change belies the more complex perturbations occurring. The increased oxidants at lower altitude and reduction at greater altitudes result in gas phase $SO_2$ oxidation increasing by 2.5-10% in the tropical and

midlatitude PBL yet decreasing at higher altitudes (Fig. 10(c,d)). This effect is expected to be even more pronounced when CS2 is compared to ST which simulates even lower low altitude OH than CS (Archer-Nicholls et al., 2020) and has been the standard mechanism for investigations into aerosol-oxidant coupling in UKCA (Thornhill et al., 2020, Weber et al., 2020, O'Connor et al., 2020). Therefore, the mechanism-driven changes to oxidants are likely to have consequences for both SOA and sulphate aerosol. While a full investigation into oxidant aerosol coupling is beyond the scope of this paper, it will form a central part of future work with the CRI mechanisms.

**5.7 Summary and synthesis**

The key changes between CS and CS2, driven by the multiple chemistry changes, can be summarised as follows:

1. $O_x$ production increases marginally in CS2 but a larger decrease in $O_x$ destruction, driven by a significant reduction in the $O(^1D) + H_2O$ flux, leads to a greater $O_3$ tropospheric burden and mixing ratios.
2. The update to the isoprene chemistry increases low altitude tropical $HO_x$ but the reduction in OH production from $O(^1D) + H_2O$ results in lower $HO_x$ concentrations in much of the free troposphere, increasing methane lifetime.
3. The update to the inorganic nitrogen reactions increases $NO_x$ as a fraction of $NO_y$ with a significant increase in the upper tropical free troposphere and a co-located increase in OH. The PAN burden decreases by 20%.
4. The increase in boundary layer oxidative capacity reduces the burden of BVOCs and confines their oxidation even more to low altitude with likely consequences for aerosol production and lifetime.

## 6. Conclusion

The radiative impact of isoprene, via its influence on atmospheric chemical composition and organic aerosol, means an accurate description of its chemical behaviour is crucial for advancing our understanding of pre-industrial, present day and future atmospheres. In this study we describe the incorporation of the Common Representative Intermediates chemistry scheme version 2.2 (CRI v2.2), along with accompanying stratospheric chemistry, into the global chemistry-climate model UKCA to create the mechanism CRI-Strat 2 (CS2). The introduction of CS2 into UKCA facilitates a semi-explicit description of $HO_x$-recycling chemistry during isoprene oxidation via the isomerisation of isoprene peroxy radicals to produce HPALDs which yield $HO_x$ upon photolysis. This is a key process for reconciling the model low bias of $HO_x$ in low $NO_x$, BVOC-rich regions. In addition, CS2 also features updates to the rate constants of the reactions of $O(^1D)$, inorganic nitrogen and organic peroxy radicals with NO and $NO_3$, bringing the mechanism into agreement with the most recent IUPAC values. CS2 is one of the first mechanisms with this functionality suitable for long term climate integrations.

A rigorous comparison using UKCA with CS2 and two other chemical mechanisms, STRAT-TROP (ST) (the standard chemistry mechanism used in UKESM1's contributions to CMIP6 experiments) and CRI-STRAT (CS) (which has tropospheric chemistry from an earlier version of the CRI, CRI v2.1), is performed against high frequency surface and airborne observational data from BVOC-rich regions for multiple chemical species including $O_3$, OH, $HO_2$, isoprene and monoterpenes and isoprene oxidation production. The $HO_x$-recycling in CS2 results in significantly enhanced

surface diel OH (up to 50% higher than CS at midday) in the Amazon and Borneo (improving model low bias), leading to improved modelling of diel and vertical isoprene profiles and reducing the mean 24-hour bias by 50-60% and 20-40% relative to ST and CS, respectively across the locations considered. However, CRI-Strat 2 exacerbates the existing isoprene model low bias away from the surface, suggesting potential issues with model vertical convection. CS and CS2 yield smaller isoprene column biases compared to observations than ST, in line with the surface and free troposphere observational comparisons, while also illustrating the significant influence the chemical mechanism has on modelled column. This comparison also highlights the significant influence the different chemistry schemes have on the simulated isoprene column and thus the considerable challenges of determining isoprene emissions via back-calculation.

The low altitude high biases for $O_3$ in CS increase modestly (1-2 ppb) in CS2. Simulated monoterpene concentrations are high biased at the surface at most of the locations considered with CS2 returning the smallest bias. As with isoprene, simulated monoterpenes display sharp morning and evening peaks which are believed to be due to boundary layer height issues. Model high bias of IEPOX and the low bias of HPALDS suggests further investigation of the key processes of loss to aerosol for IEPOX and HPALD photolysis frequency are needed.

In addition to observational comparisons, a detailed comparison of UKCA model output using CS2 is performed, complementing the earlier comparison of ST and CS (Archer-Nicholls et al., 2020). Sensitivity tests are also performed to help isolate the drivers of the differences between CS and CS2. CS2 simulates an 8% increase in tropospheric $O_3$ burden driven primarily by reduced $O_x$ loss as the changes to rate constants of $O(^1D)$ with $H_2O$, $O_2$ and $N_2$ mean that a smaller fraction of $O(^1D)$ reacts with $H_2O$ to produce OH. Low altitude $O_3$ increased by 2-4 ppb over much of the globe, driven predominantly by changes to the $O(^1D)$ and inorganic nitrogen reactions. More broadly, the widespread influence of the changes the rate constants of $O(^1D)$ and multiple inorganic nitrogen species highlights the importance of having accurate information for these parameters.

Relative to CS, low altitude OH increased over terrestrial regions, exceeding 50% in some tropical forested regions, primarily due to the influence of $HO_x$-recycling from isoprene. However, OH decreased over the oceans and in much of the free troposphere driven by updates to the rate constants of $O(^1D)$'s reactions with $H_2O$, $O_2$ and $N_2$. As a result, methane lifetime increased by 1.9%, in stark contrast to previous studies using CRI v2.2 in the STOCHEM model which did not make changes to $O(^1D)$ and inorganic nitrogen reactions. When the changes to isoprene chemistry were isolated, methane lifetime decreased by 2.2%, qualitatively in line with previous studies. The addition of isomerisation pathways in the updated isoprene scheme reduced the methyl (7%) and non-methyl peroxy (36%) radical burdens.

The distribution of nitrated species ($NO_y$) in CS2 was closer to that simulated in ST than CS with a significant reduction (20%) in the burden of PANs which was driven by a reduction in the precursor $RO_2$. The $NO_x$ burden increased by 4%.

The increase in low altitude OH reduced the burdens of isoprene (25%) and monoterpenes (11-18%) and the extent of their dispersion: more oxidation took place in the boundary layer where loss of oxidation products such as the lumped SOA precursor Sec_Org to existing aerosol is likely to be greater. Enhanced $SO_2$ oxidation in the boundary layer was also simulated. These changes are likely to have implications for SOA and sulphate aerosol, particularly as CS has already been shown to have a more highly oxidising boundary layer than ST. Therefore, the difference between CS2 and ST (the mechanism used to explore chemical-aerosol coupling in UKESM1 in CMIP6 experiments), is likely to be significant and will be the subject of future work.

The addition of CS2 also lays the groundwork for the incorporation of a novel chemistry scheme which describes the formation of the highly oxidised organic molecules (HOMs) derived from biogenic species such as α-pinene (e.g. CRI-HOM, Weber et al., 2020b). HOMs are crucial for new particle formation without sulphuric acid (Kirkby et al., 2016, Simon et al., 2020), a process which is an important source of new particles in the Amazonian free troposphere (Zhao et al., 2020) and has been simulated to have consequences for our understanding of pre-industrial aerosol burden (Gordon et al., 2016). The influence of isoprene in HOM production (Kiendler-Schaar et al., 2009, McFiggans et al., 2019, Heinritzi et al., 2020) can also be captured by addition of CRI-HOM making UKCA one of the very first global chemistry-climate models to feature a semi-explicit representation of HOMs and enabling further investigation into the climatic impact of the interaction between BVOCS. Long chain terpenes addition to CS2 are also planned including sesquiterpenes, which may reduce the surface ozone high bias and form HOMs, and improvements to the uptake of oxidised species to plant surfaces.

While certain elements of the CRI-STRAT 2 mechanism in UKCA such as the ozone high bias remain problematic, its incorporation represents a major step forward in our ability to simulate isoprene chemistry in low $NO_x$ environments. The simulated changes to oxidants in CRI-Strat 2 will affect the atmosphere's radiative balance by perturbing certain greenhouse gases and aerosols and investigating the impact will be a major topic of future work. In particular, the feedback between the biosphere and climate, mediated by BVOCS, will be evaluated using multiple mechanisms to assess their influence. CRI-Strat 2 can be taken up for use, alongside other mechanisms, to further our understanding of the wide-ranging impact BVOCs have on climate.

*Data Availability:*

The description of the Z2F field campaign is given SI Section S4 and the observational data is available at https://doi.org/10.17863/CAM.65133

The observational data from the SEAC4RS flight campaign is available at https://www-air.larc.nasa.gov/cgi-bin/ArcView/seac4rs?MERGE=1#60_SECOND.DC8_MRG/.

The observational data from the ATTO tower is available to download at https://www.attodata.org. Specific datasets used were https://www.attodata.org/ddm/data/Showdata/72, https://www.attodata.org/ddm/data/Showdata/73, https://www.attodata.org/ddm/data/Showdata/74 and https://www.attodata.org/ddm/data/Showdata/77.

The observational data from the FAAM aircraft is available at http://data.ceda.ac.uk/badc/op3/data/op3-aircraft and Borneo data can be found at http://data.ceda.ac.uk/badc/op3/data.

Data tables of the full CRI-Strat 2 mechanism and the mechanisms used in the sensitivity test described in this paper are included in the supplement. The CRI v2.2 mechanism can be viewed and downloaded from http://cri.york.ac.uk.

Model data and analysis code is available from JW on request.

*Code Availability*

Due to intellectual property right restrictions, we cannot provide either the source code or documentation papers for the UM. The Met Office United Model is available for use under licence. A number of research organisations and national meteorological services use the UM in collaboration with the UK Met Office to undertake basic atmospheric process research, produce forecasts, develop the UM code and build and evaluate Earth system models. For further information on how to apply for a licence; see https://www.metoffice.gov.uk/research/approach/modelling-systems/unified-model (last access: 24 November 2020).

*Author Contributions*

Mechanism incorporation was carried out by JW, SAN and NLA with advice from ATA, YMS, MJ, MAHK and DES. Observational comparison experiments were designed by JW, SAN, NLA and ATA and executed by JW, mechanism-mechanism comparison experiments were designed by JW, ATA, NLA and SAN and executed by JW. TJB, CJP, AB and PA compiled and supplied the Z2F Brazil observational data and TJB wrote the field campaign description in the SI, RS advised on the SEAC$^4$RS data and analysis, JW, SAN, ATA, JW* interpreted the Z2F Brazil, Borneo, ATTO, FAAM, GABRIEL and SEAC$^4$RS observational data. JW wrote the paper. All co-authors discussed the results and commented on the paper.

(JW = James Weber, JW* = Jonathan Williams)

*Competing interests.*

The authors declare that they have no conflict of interest.

*Financial support.*

JMW has been funded by a Vice-Chancellor's Award from the Cambridge Trust. SAN and ATA have been funded by NERC PROMOTE (grant no. NE/P016383/1). NLA and ATA are supported by NERC and NCAS through the

ACSIS project. YMS has been funded by NERC through the University of Cambridge ESS-DTP. MAHK and DES are funded by NERC (grant code-NE/K004905/1), Bristol ChemLabS and the Primary Science Teaching Trust. TB, CJP, AB and PA acknowledge funding from FAPESP – Fundação de Amparo à Pesquisa do Estado de São Paulo, grant number 2017/17047-0.

*Acknowledgements*

This work used Monsoon2, a collaborative High Performance Computing facility funded by the Met Office and the Natural Environment Research Council. This work used JASMIN, the UK collaborative data analysis facility.

We are grateful to Dr Horst Fischer, Dr Hartwig Harder, Dr Pete Edwards for their assistance and advice and to Professor Jason Surratt for providing the calibration standards for the Z2F Brazil study. We thank the field support from the LBA central office at INPA in Manaus.

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

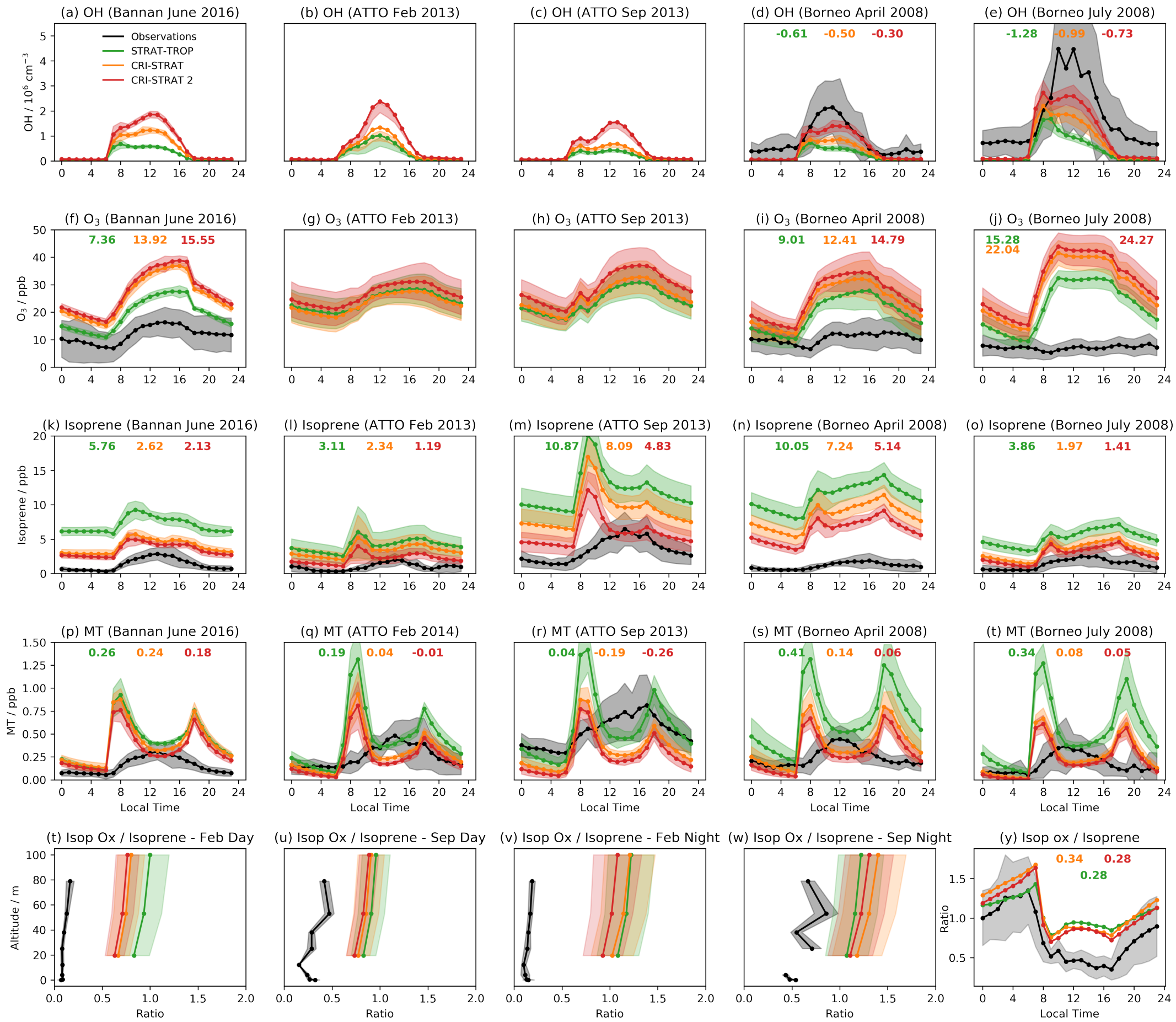

**Figure 1. Mean diel cycles of observed and modelled OH (top row), $O_3$ (2nd row), Isoprene (3rd row) and MT (MT=α-pinene + β-pinene for the CRI mechanisms) (4th row) at the three surface/near surface sites considered. The bottom row shows the vertical profile of the ratio of isoprene oxidation products to isoprene for daytime (0900-1500 LT) and nighttime (2100-0300 LT) periods and the diel profile of the ratio at 53 m (all from ATTO tower). Shading indicates ±1 standard deviation from the mean and the numbers in bold show the mean diel model bias (model - observations) for species/locations where observations were recorded.**

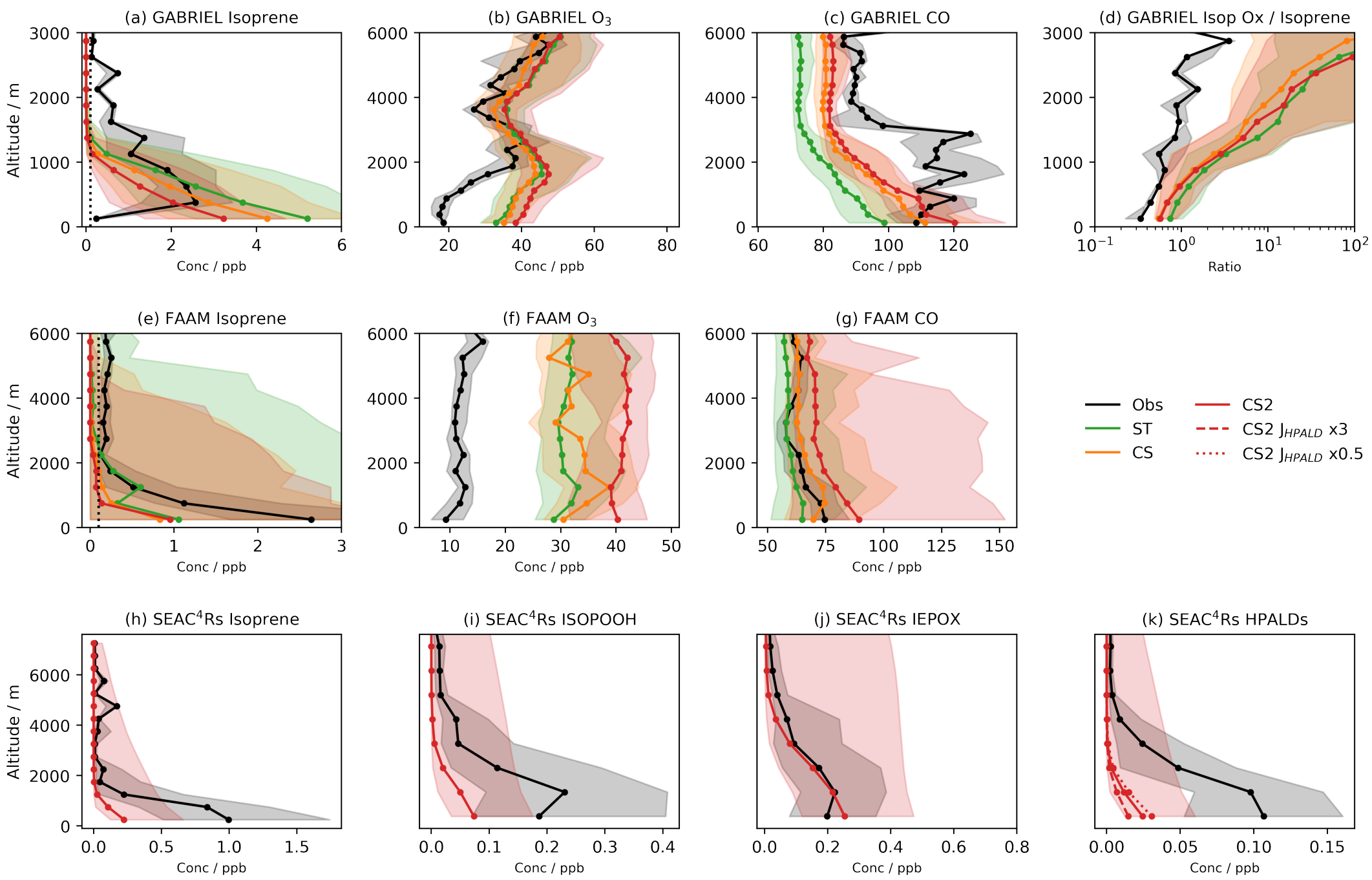

**Figure 2. Median observed and model concentrations for the GABRIEL campaign in the Amazon for (a) Isoprene, (b) $O_3$, (c) CO and (d) the ratio of isoprene oxidation products to isoprene. Median observed and model concentrations for the FAAM campaign over Borneo for (e) isoprene, (f) $O_3$ and (g) CO. Median observed and model concentrations for the SEAC$^4$RS campaign over the South East USA for (h) isoprene, (i) isoprene hydroperoxide (ISOPOOH), (j) the isoprene epoxy diol (IEPOX) and (k) hydroperoxy aldehydes (HPALDs). SEAC$^4$RS observational data is also filtered to exclude urban plumes ($NO_2$>4 ppb), fire plumes (acetonitrile>0.2 ppb) and stratospheric air ($O_3$/CO > 1.25) as done in Schwantes et al (2020). Shading shows IQR, black dotted lines (a, e) show estimated limits of detection for isoprene and $J_{HPALD}$x3 and $J_{HPALD}$x0.5 lines in (k) show results of the scaling the HPALD photolysis frequency by 3 and 0.5, respectively. Note the logarithmic horizontal scale for (d).**

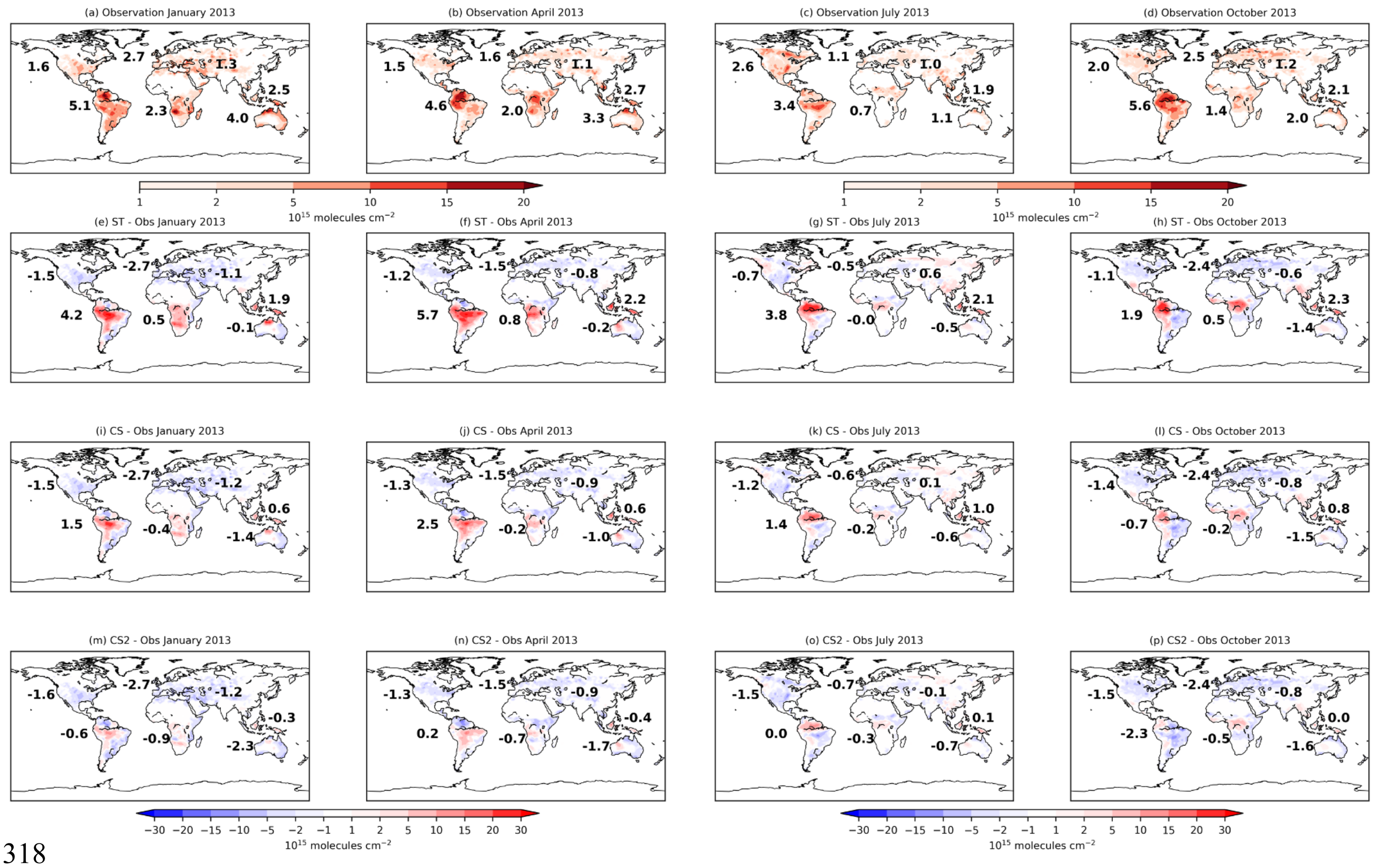

**Figure 3. Monthly mean isoprene column values from the Global Isoprene Column observational dataset (Wells et al., 2020) for (a) January, (b) April, (c) July and (d) October 2013. Model bias (model-observation) using (d-h) ST (i-l) CS, and (m-p) CS2. Numbers in (a-d) show area-weighted mean model column values and in (e-p) model bias for individual terrestrial regions (number in North Atlantic refers to Europe and South Atlantic to Africa).**

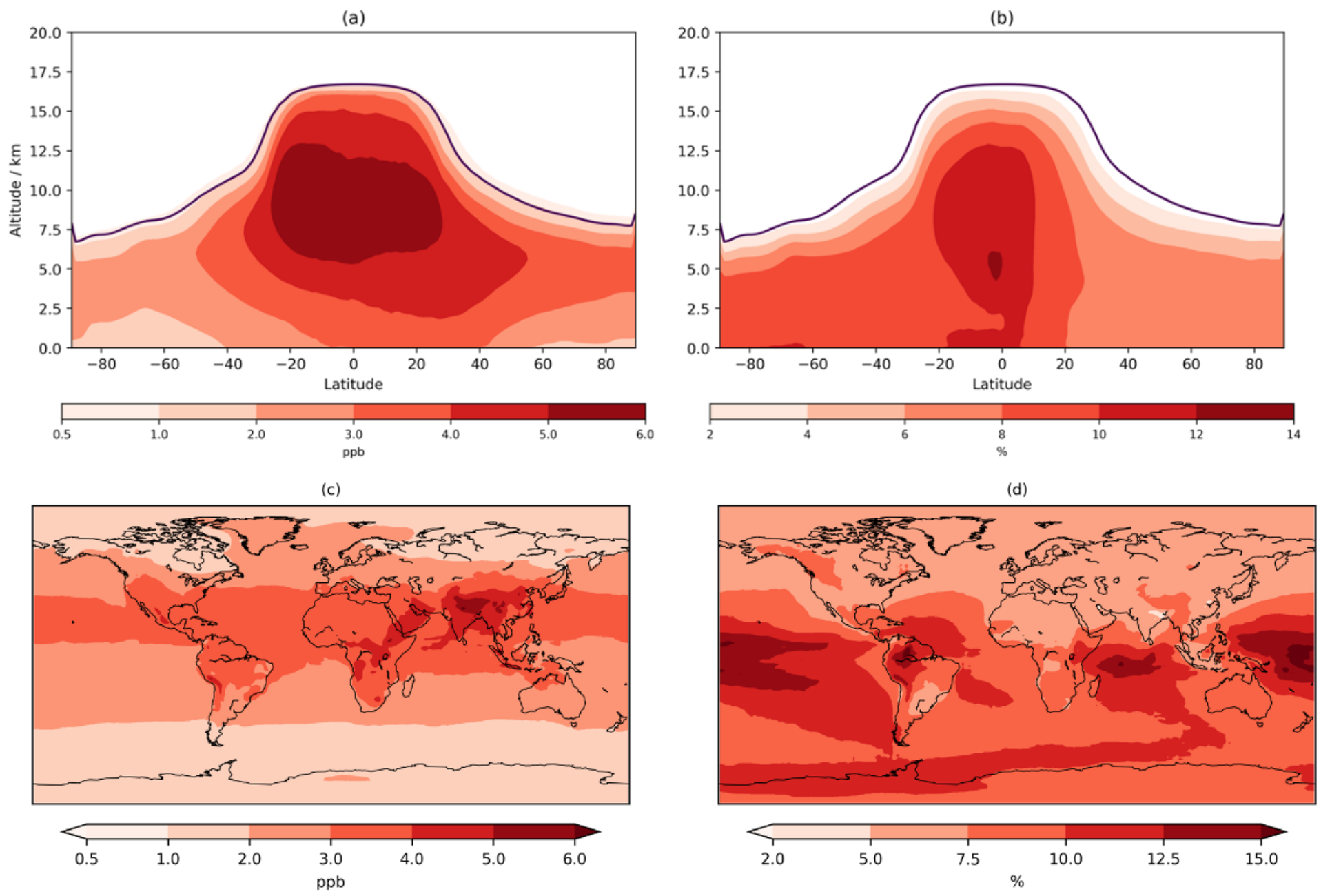

**Figure 4. Annual mean tropospheric zonal (a, b) and lowest 500 m (c, d) change in $O_3$ mixing ratio (CS2 - CS). Purple line in zonal mean shows average height of tropopause.**

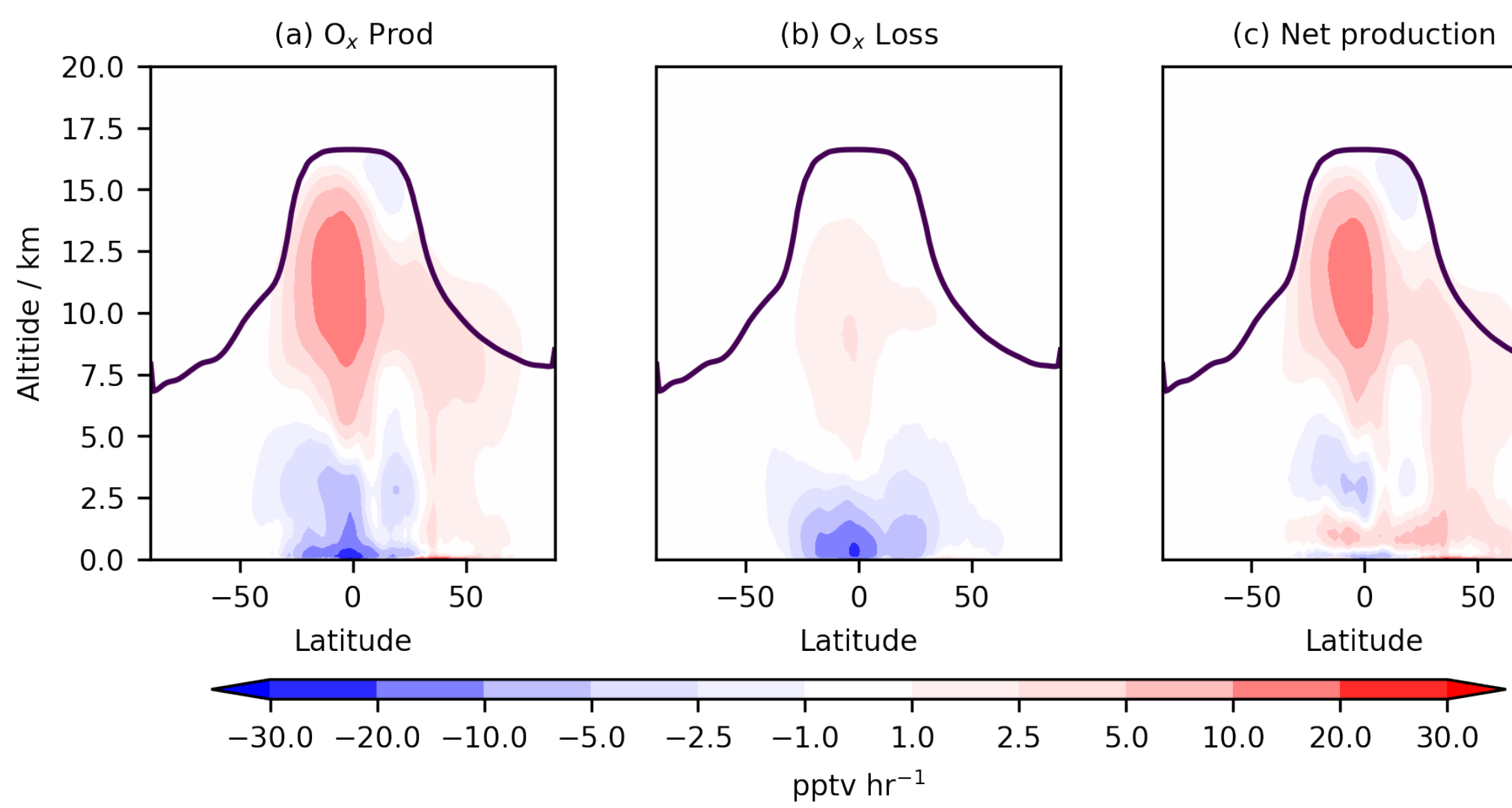

**Figure 5. Annual zonal mean change in (a) total $O_x$ production flux, (b) total $O_x$ chemical loss flux and (c) net $O_x$ chemical production flux. Purple line indicates mean tropopause height.**

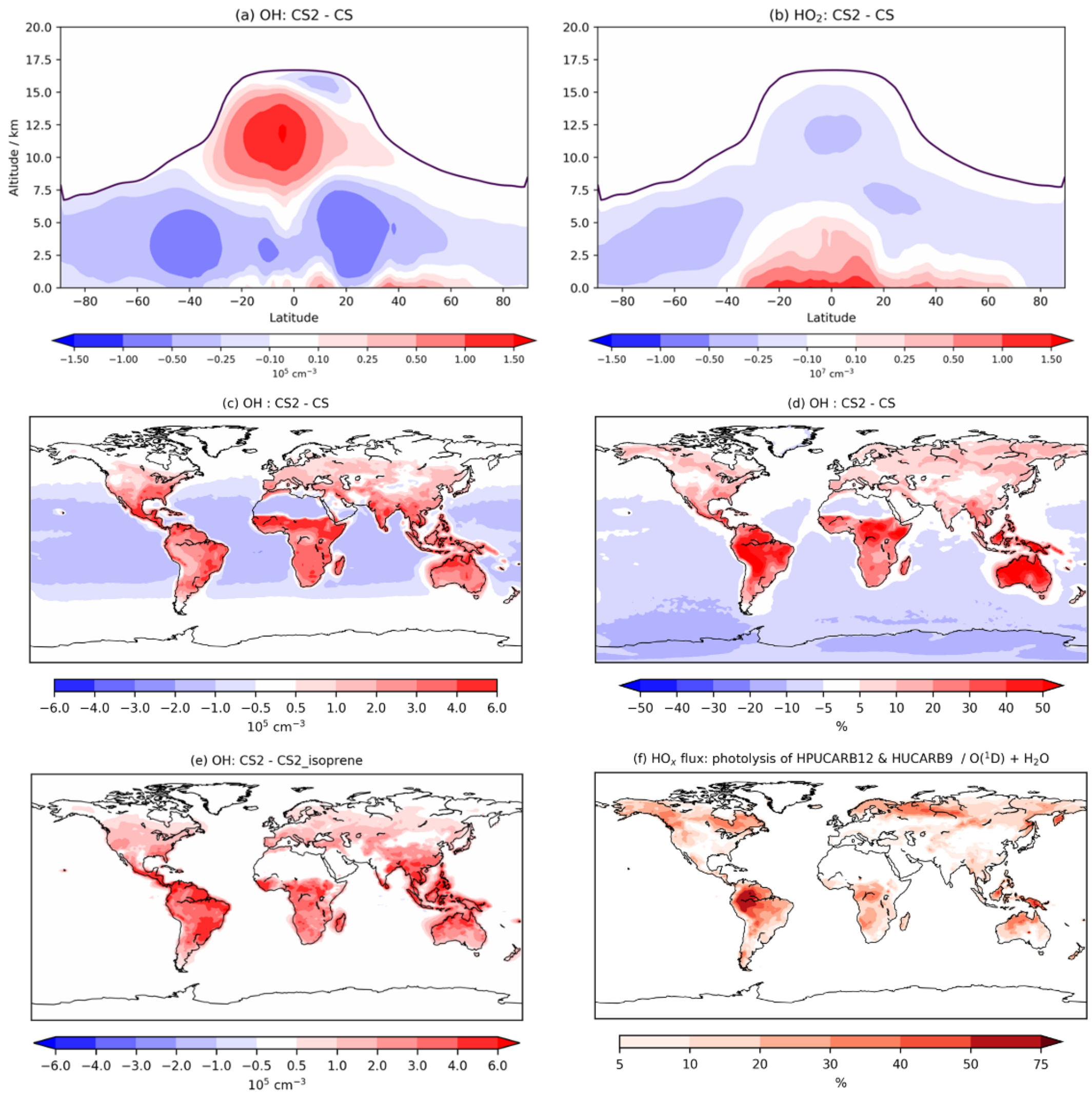

**Figure 6. Annual zonal mean changes in (a) OH and (b) $HO_2$ between CS2 and CS, (c) absolute and (d) percentage in change in OH in lowest ~500 of atmosphere, (e) the change in OH in lowest 500 m between the CS2 and CS2_isoprene sensitivity test and (f) $HO_x$ production flux from HPUCARB12 and HUCARB9 photolysis as a percentage of $HO_x$ from $O(^1D)$ + $H_2O$ (right, bottom). Purple lines indicate average height of tropopause.**

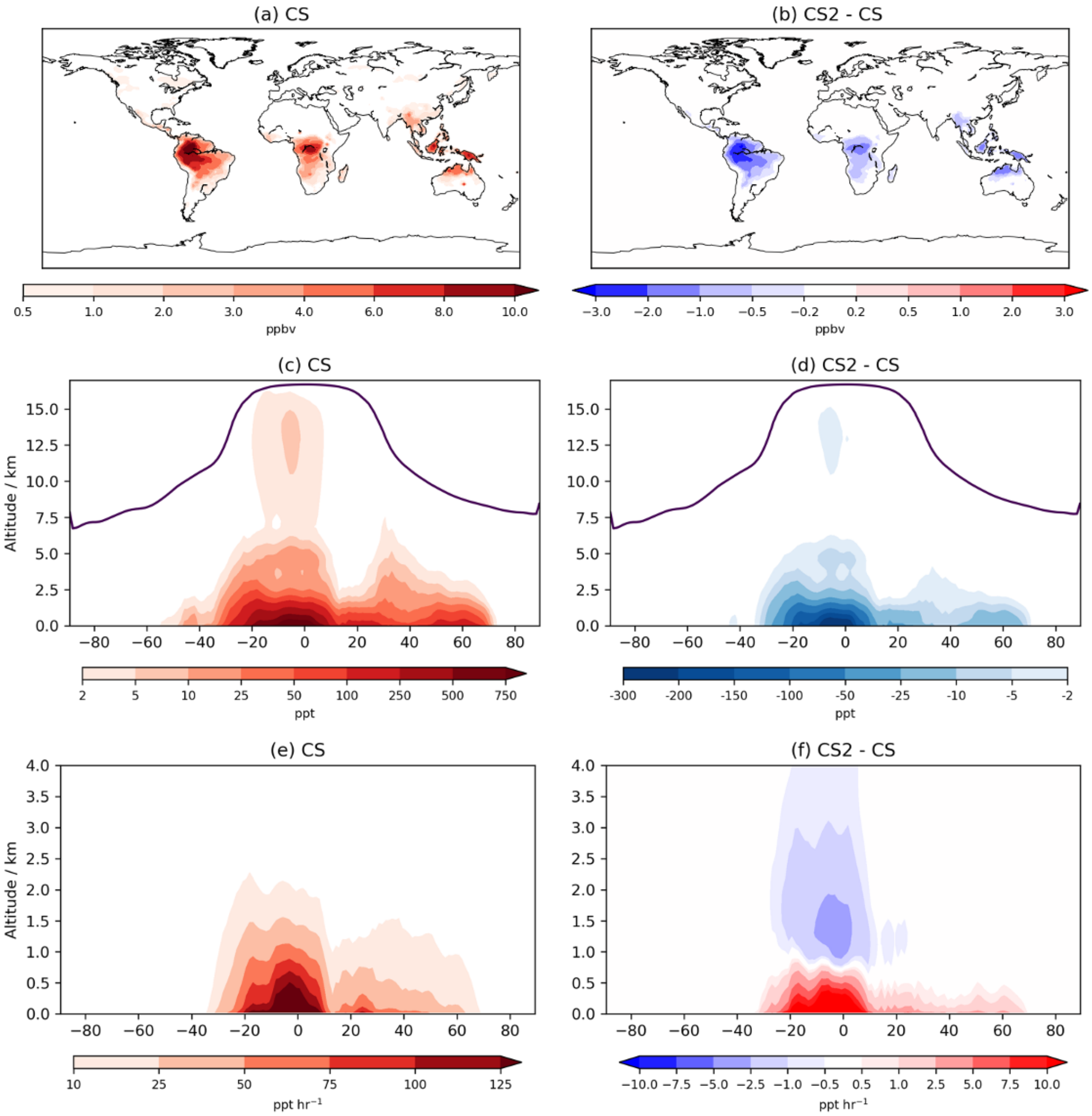

**Figure 7. Annual mixing ratio of isoprene averaged over the lowest ~ 100 m (a) in CS and (b) the difference between CS2 and CS. Annual zonal mean mixing ratios in (c) CS and (d) difference between CS2 and CS (note the log scales). Annual average total oxidation flux of isoprene (e) in CS and (f) the difference between CS2 and CS.**

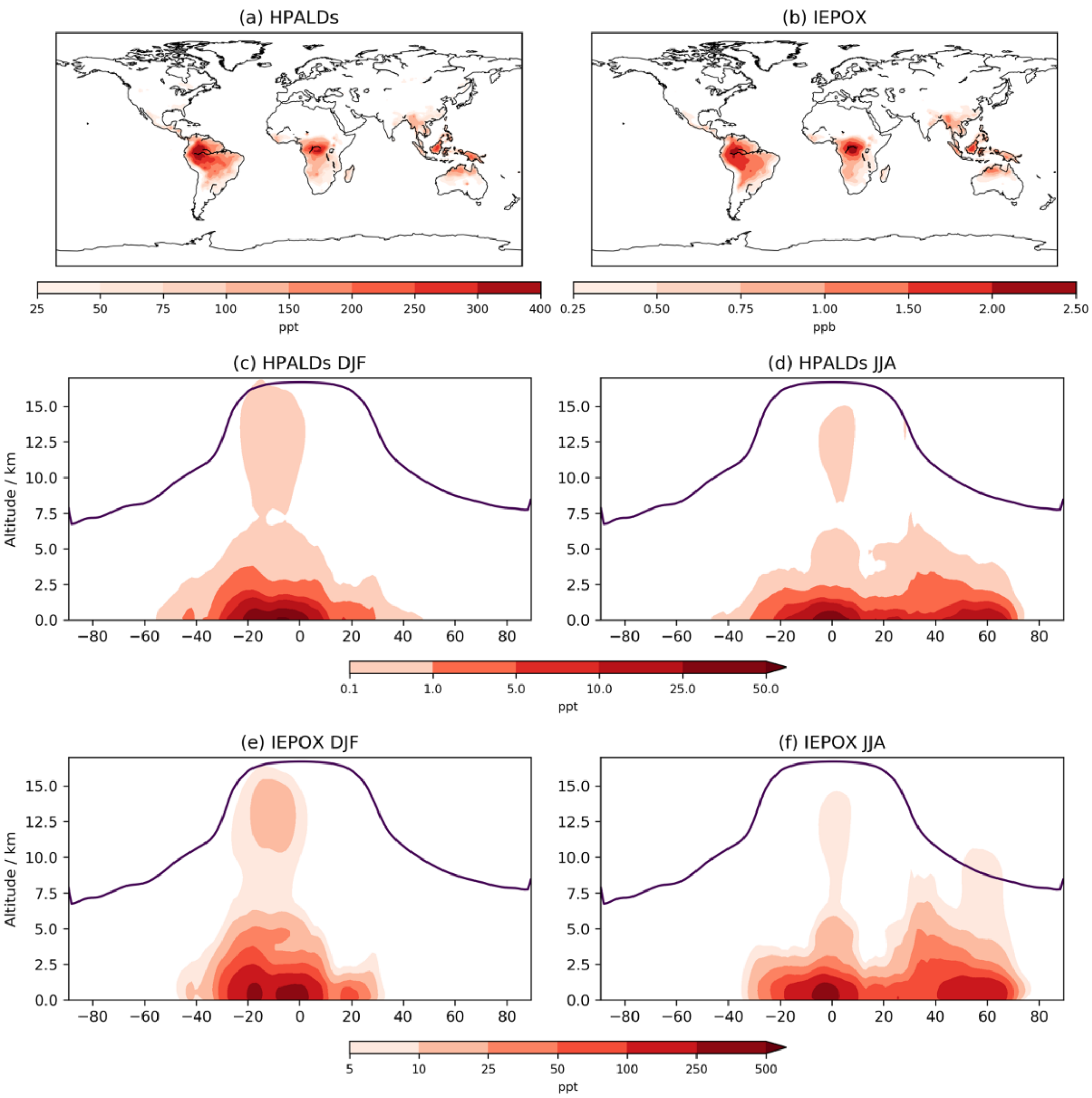

1356 **Figure 8. Annual mean mixing ratios for (a) HPALDs and (b) IEPOX (upper panels) over lowest ~100 m. DJF and JJA zonal mean mixing ratios for HPALDs (c, d) and IEPOX (e,f), note differing scales for HPALD and IEPOX plots and log scales for (c-f).**

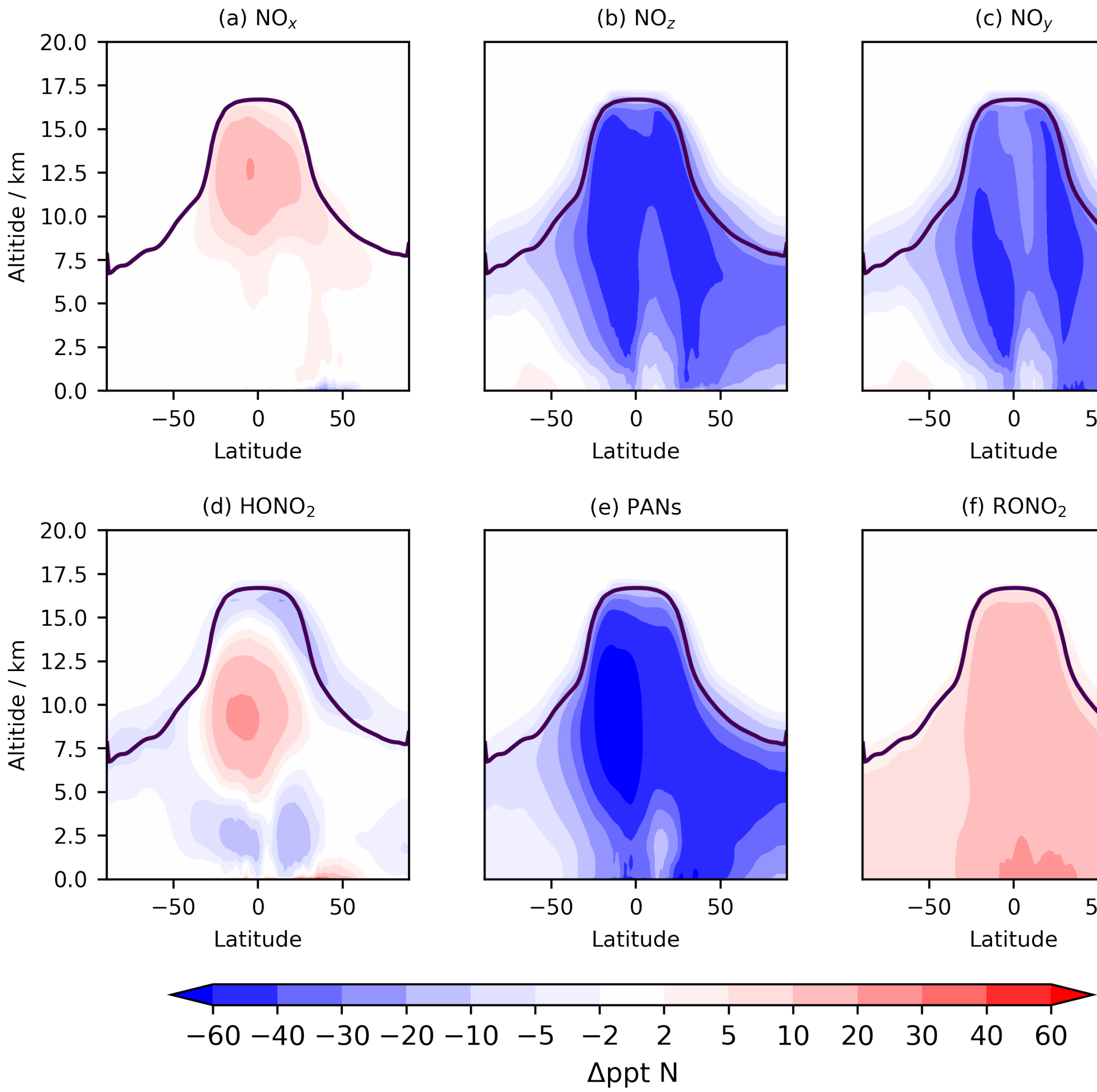

**Figure 9. Tropospheric annual zonal mean change in (a) $NO_x$, (b) $NO_z$, (c) $NO_y$, (d) $HONO_2$, (e) PANs and (d) $RONO_2$ between CS2 and CS. Purple line shows average tropopause height.**

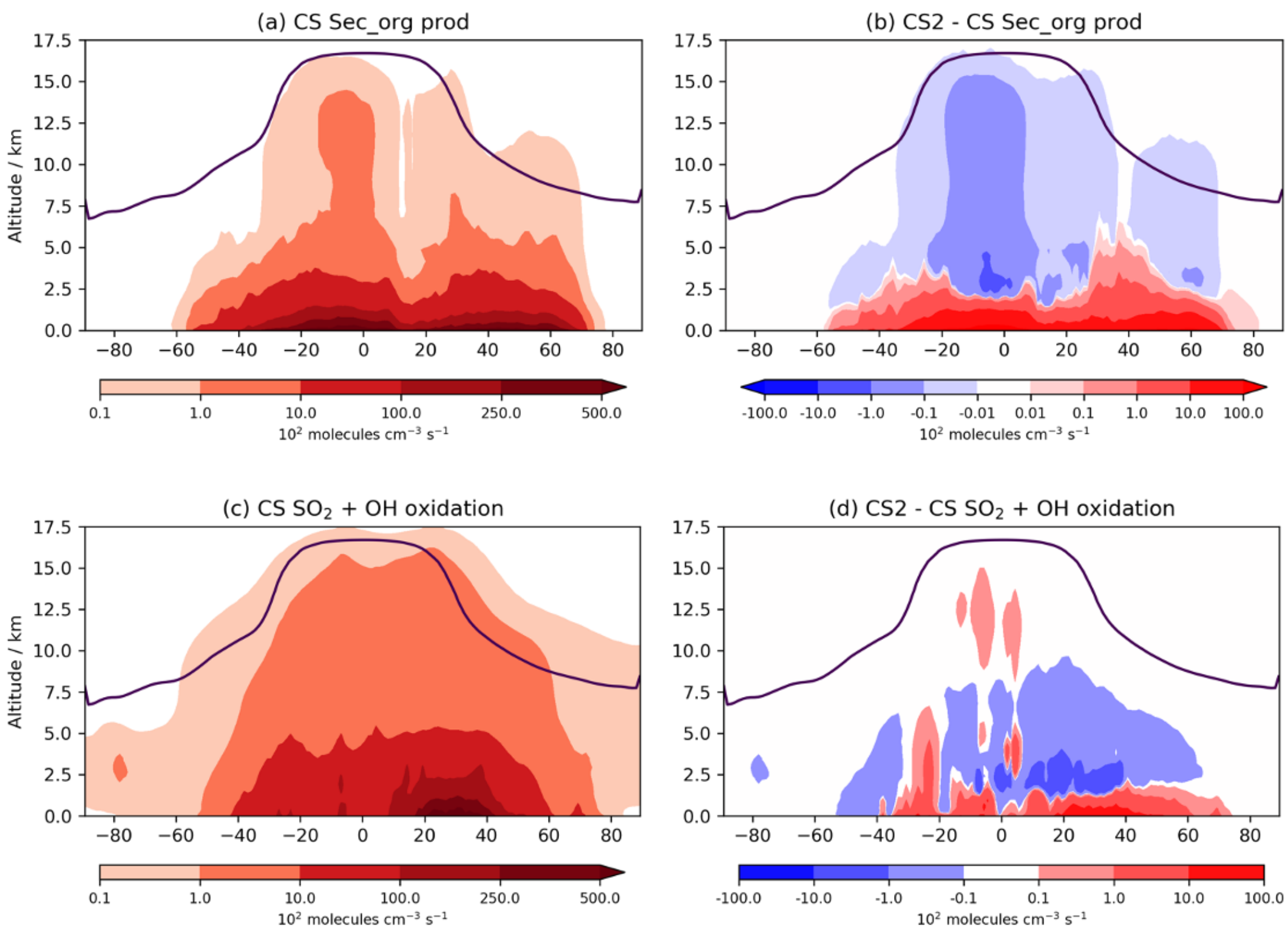

**Figure 10. Tropospheric annual zonal mean production flux of Sec_Org in (a) CS and (b) difference between CS2 and CS. Annual zonal mean flux of $SO_2$ + OH in (c) CS and (d) difference between CS2 and CS.**

**Table 1. Comparison of the CRI-STRAT and CRI-STRAT 2 chemical mechanisms**

| | **CRI-STRAT (CS)** | **CRI-STRAT 2 (CS2)** |
|---|---|---|
| **Tropospheric Chemistry Scheme** | CRI v2.1 (Jenkin et al., 2008, Watson et al., 2008, Utembe et al., 2010) | CRI v2.2 (Jenkin et al., 2019) |
| **Stratospheric Chemistry Scheme** | Stratospheric chemistry (Morgenstern et al., 2009; Archibald et al., 2020) | Stratospheric chemistry (Morgenstern et al., 2009; Archibald et al., 2020) |
| **No. of Species** | 219 | 228 |

Open Access EGU

| | | |
|---|---|---|
| **No. of Bimolecular Reactions** | 536 | 582 |
| **No. of Termolecular Reactions** | 36 | 44 |
| **No. of Photolysis Reactions** | 128 | 140 |

**Table 2. Species added and removed from the CS mechanism in the development of the CRI-Strat 2 mechanism.**

| **Added Species** | **Species Functionality** | **MCM v3.3.1 equivalent** |
|---|---|---|
| HPUCARB12 | Hydroperoxy aldehyde (HPALD) | C5HPALD1, C5HPALD2 |
| HUCARB9 | Unsaturated hydroxy carbonyl | HMVK, HMAC |
| IEPOX | Isoprene epoxy diol | IEPOXA, IEPOXB, IEPOXC |
| HMML | Hydroxymethyl-methyl-a-lactone | HMML |
| DHPCARB9 | Dihydroperoxy carbonyl | DHPMEK, DHPMPAL |
| DHPR12OOH | Trihydroperoxy carbonyl | C536OOH |
| DHCARB9 | Dihydroxy carbonyl | HO12CO3C4 |
| RU12NO3 | Hydroxy carbonyl nitrate | C57NO3, C58NO3, C58ANO3 |
| RU10NO3 | Hydroxy carbonyl nitrate | MVKNO3, MACRNO3 |
| DHPR12O2 | Dihydroperoxy carbonyl peroxy radical | C536O2, C537O2 |
| MACO3 | Unsaturated acyl peroxy radical | MACO3 |
| RU10AO2 | Hydroxy carbonyl peroxy radical | MACRO2 |

| **Removed Species** | | **MCM v3.2 equivalent** |
|---|---|---|
| RU12PAN | PAN-type species with at least one hydroxy group | C5PAN19 |
| TNCARB11 | Alkyl carbonyl | N/A |
| TNCARB12 | Alkyl carbonyl | N/A |

**Table 3 - Shorter runs performed for mechanism-observation comparisons. Identical biogenic (2001-2010 MEGAN-MACC climatology, iBVOC for isoprene and MT) and ocean (1990 timeslice) for each run unless otherwise stated.**

| **Run Name** | **Mechanisms Tested** | **Period(s)** | **Observational Reference** |
|---|---|---|---|
| ATTO | ST, CS, CS2 | Feb 2013, Sept 2013, Feb 2014 | Yanez-Serrano et al (2015) |
| ZF2 Brazil | ST, CS, CS2 | June 2016 | See SI Section S4 |
| Borneo | ST, CS, CS2 | April-May, June-July 2008 | Hewitt et al (2010), Whalley et al (2011), Edwards et al (2013) |
| GABRIEL | ST, CS, CS2 | October 2005 | Butler et al (2008) |
| FAAM | ST, CS, CS2 | July 2008 | Hewitt et al (2010) |
| Isoprene Column | ST, CS, CS2 | Jan, April, Jul & Oct 2013 | Wells et al (2020) |
| SEAC$^4$RS | CS2 | August-September 2013 | Toon et al (2016) |

**Table 4 - Longer runs performed for CRI mechanism comparison. Identical emissions for each run (anthropogenic and biomass timeslice 2014, biogenic 2001-2010 MEGAN-MACC climatology, oceanic 1990 timeslice)**

| Name | Base Mechanism | Total Length and Period | Alterations from base mechanism |
|---|---|---|---|
| CS | CRI-STRAT | 5 years (1 year spin up) | None |
| CS2 | CRI-STRAT 2 | 5 years (1 year spin up) | None |
| CS2_O1D | CS2 | 2 years (1 year spin up) | Rate constants for O($^1$D) with $H_2O$, $O_2$ and $N_2$ set to values in CS |
| CS2_inorgN | CS2 | 2 years (1 year spin up) | Rate constants for $HONO_2$, $HO_2NO_2$, $N_2O_5$, PAN formation, $HO_2$ + NO and $MeONO_2$ + OH set to values in CS |
| CS2_isoprene | CS2 | 2 years (1 year spin up) | Isoprene chemistry set to that in CS |
| CS2_RO2_N | CS2 | 2 years (1 year spin up) | Rate constants for $RO_2$ + NO and $RO_2$ + $NO_3$ reactions reverted to CS values |
| CS2_photo (see SI Section 6) | CS2 | 2 years (1 year spin up) | Photolysis of CARB3, HCHO and EtCHO reverted to that from CS |

**Table 5. Location, reference, time period and species measured in observational data sets and corresponding modelling approach. For the Z2F Brazil, ATTO, Borneo, GABRIEL, FAAM and SE4C$^4$RS datasets, model data was filtered to select only the same days as observational data.**

| Dataset | Reference | Dates of | Measurement | Species | Corresponding |
|---|---|---|---|---|---|

| **(Location / Coordinates)** | | **Measurement** | **Details** | **Considered** | **model run (Table 3 unless stated)** |
|---|---|---|---|---|---|
| ZF2 Brazil Field Campaign, Amazon (-2.60°, -60.21°, 60 km NNW of Manaus) | See SI Section S4 | 22 June 2016 - 5 July 2016 | 1-minute interval measurements 30 m above ground (above tree canopy) | $O_3$, CO, $SO_2$, $NO_2$, isoprene, monoterpenes, benzene | ZF2 Brazil |
| Instant ATTO Tower, Amazon (-2.14° , -59.00°, 150 km NE of Manaus) | Yannez-Serrano et al (2015) | February 2013, September 2013 and February 2014 | 16-minute interval measurements at multiple heights above ground (0.05 m, 0.5 m, 4 m, 12 m, 25 m, 38m, 53m and 79 m) | Isoprene, monoterpenes, methyl vinyl ketone (MVK), methacrolein (MACR), isoprene hydroperoxide (ISOPOOH), acetone<br><br>(All PTRMS) | ATTO |
| GAW Station, Borneo (5.0°, 117.5°) | Hewitt et al (2010), Whalley et al (2011), Edwards et al (2013) | April-July 2008 | 10-minute intervals | OH, $HO_2$ (both FAGE), $O_3$ (Thermo Electron Instrument ) isoprene, monoterpene (both PTRMS), HCHO (aerolaser Hantzsch), CO (Aerolaser AL5002), MeCHO, acetone MACR, | Borneo |

| | | | | | |
|---|---|---|---|---|---|
| | | | | MVK (both GC-FID), PAN (GC-MS), $NO_2$ (Thermo Environmental Instruments 42C) | |
| GABRIEL Aircraft Campaign (Suriname, Guyana and French Guiana) | Butler et al (2008) | October 2005 | Daytime aircraft measurements sampling ~0.3-8 km at 30 second intervals | $O_3$, NO (both ECOEX), HCHO, CO (both MPIC TRISTAR), acetone, isoprene, MACR, MVK (all PTRMS) | GABRIEL |
| FAAM Aircraft Campaign, Borneo | Hewitt et al (2010) | July 2008 | Daytime aircraft measurements sampling ~0.3-7 km at 5 min intervals | $O_3$ (TECO 49), isoprene (PTRMS), CO (AERO AL5002) | FAAM |
| SE4C$^4$RS Flight Campaign (Southeast United States) | Toon et al (2016) | August - September 2013 | Daytime aircraft measurements sampling up to 12 km at 1 min intervals | $O_3$ (ERSL), CO (DACOM), Isoprene (WAS), ISOPOOH, HPALDs, IEPOX, isoprene nitrate (all CIT) | SEAC$^4$RS |
| Global Isoprene Columns | Wells et al (2020) | Jan, April, Jul & Oct 2013 | Global monthly mean isoprene column values | Isoprene | Isoprene Column |

**Table 6 - Annual mean $O_x$ diagnostics for CRI-STRAT, CRI-STRAT 2 and difference between mechanisms (percentage changes in parentheses). UKESM1 CMIP6 1995-2004 using ST: chemical production = 5315 Tg year$^{-1}$, chemical loss = 4476 Tg year$^{-1}$, dry deposition = 867 Tg year$^{-1}$ (Griffiths et al., 2021)**

| | **CS** | **CS2** | **CS2 - CS** |
|---|---|---|---|
| $O_3$ Burden (Tg) | 328 | 354 | 26 (7.9%) |
| $O_x$ Lifetime (days) | 17.4 | 18.8 | 1.4 (8.0%) |
| OPE | 33.74 | 33.78 | 0.05 (0.1%) |
| **Chemical Production (Tg year$^{-1}$)** | **6572** | **6582** | **10 (0.1%)** |
| $HO_2$ + NO | 4099 | 4322 | 132 (3.2%) |
| MeOO + NO | 1573 | 1583 | 10 (0.6%) |
| NO + $RO_2$ | 849 | 717 | -131 (-15.4%) |
| Other | 51 | 49 | -1 (-2.8%) |
| **Chemical Loss (Tg year$^{-1}$)** | **5834** | **5757** | **-77 (1.3%)** |
| $O(^1D) + H_2O$ | 3157 | 2928 | -229 (-7.2%) |
| $HO_2 + O_3$ | 1666 | 1819 | 152 (9.1%) |
| $OH + O_3$ | 740 | 796 | 57 (7.6%) |
| $O_3$ + Alkene | 166 | 101 | -65 (-39.2%) |
| Other | 105 | 113 | 8 (10.1%) |
| **Deposition (Tg year$^{-1}$)** | **1133** | **1207** | **76 (6.5%)** |
| $O_3$ Dry Dep | 942 | 1018 | 77 (8.0%) |
| $NO_y$ dep | 191 | 189 | -3 (1.3%) |
| Inferred STT (Tg year$^{-1}$) | 395 | 384 | -13 (-3.3%) |

**Table 7 – Tropospheric average $HO_x$ parameters for CS and CS2.**

| | CS | CS2 | CS2 – CS |
|---|---|---|---|
| [OH] / $10^6$ $cm^{-3}$ | 1.355 | 1.334 | -0.021 (1.5%) |
| [$HO_2$] / $10^8$ $cm^{-3}$ | 0.990 | 0.988 | -0.002 (0.2%) |
| [OH] / [$HO_2$] (%) | 1.369 | 1.349 | -0.02 (1.5%) |
| $CH_4$ lifetime w.r.t. OH / years | 7.43 | 7.60 | -0.17 (2.3%) |

**Table 8 - Burdens of $NO_y$ and its constituent species, $NO_x$ emissions, $NO_y$ deposition and inferred Stratosphere-to-Troposphere (STT) transport of $NO_y$. Values in parentheses for burdens show the fraction of total $NO_y$ burden represented by each constituent and, for deposition diagnostics, the fraction of total $NO_y$ deposition represented by each pathway.**

| | CS | CS2 | CS2 – CS |
|---|---|---|---|
| **$NO_y$ burden / TgN** | **1.088** | **1.036** | **-0.052** |
| $NO_x$ burden / TgN | 0.118 (10.9%) | 0.123 (11.9%) | 0.005 |
| $NO_z$ burden / TgN | 0.972 (89.2%) | 0.914 (88.1%) | -0.058 |
| $HONO_2$ burden / TgN | 0.523 (48.0%) | 0.521 (50.3%) | -0.002 |
| Other inorganic $NO_z$ burden / TgN | 0.020 (1.8%) | 0.014 (1.4%) | -0.006 |
| PANs burden / TgN | 0.367 (33.7%) | 0.292 (28.2%) | -0.075 |
| $RONO_2$ burden / TgN | 0.044 (4.0%) | 0.070 (6.7%) | 0.026 |
| $MeO_2NO_2$ burden / TgN | 0.008 (0.8%) | 0.008 (0.7%) | -0.0007 |

| | | | |
|---|---|---|---|
| Nitrophenols burden / TgN | 0.009 (0.9%) | 0.009 (0.9%) | -0.0005 |
| **NOx Emissions / TgN year$^{-1}$** | **55.65** | **55.65** | **0** |
| **Total NOy Deposition / TgN year$^{-1}$** | **62.12** | **62.35** | **0.23** |
| **Inferred STT / TgN year$^{-1}$** | **6.47** | **6.70** | **0.23** |
| $NO_x$ Deposition / TgN year$^{-1}$ | 6.32 (10.2 %) | 6.30 (10.1 %) | -0.02 |
| $HONO_2$ Wet Deposition / TgN year$^{-1}$ | 29.01 (46.6%) | 29.26 (46.8 %) | 0.25 |
| $HONO_2$ Dry Deposition / TgN year$^{-1}$ | 21.66 (34.9 %) | 21.79 (35.0 %) | 0.13 |
| Other Inorganic NOy Deposition / TgN year$^{-1}$ | 1.21 (2.0 %) | 0.96 (1.5 %) | -0.25 |
| PANs / TgN year$^{-1}$ | 2.45 (3.9%) | 1.93 (3.1 %) | -0.52 |
| $RONO_2$ Deposition / TgN year$^{-1}$ | 1.41 (2.3 %) | 2.03 (3.2 % ) | 0.62 |
| Nitrophenols Deposition / TgN year$^{-1}$ | 0.08 (0.1 %) | 0.07 (0.1 %) | -0.01 |