# Peer review of "Improvements to the representation of BVOC chemistry-climate interactions in UKCA (vn11.5) with the CRI-Strat 2 mechanism: Incorporation and Evaluation"

_Geoscientific Model Development, 2021_

## Author Response (AR1)

**Response to Reviewers**

We are very grateful to both reviewers for their comments and efforts which have helped us improve this manuscript. Following the recommended structure, we have responded to each reviewers' comments sequentially below with italicised text showing the reviewer's comments and plain text showing our response. Text which has been added to the manuscript is coloured red. Original manuscript text is in blue and any text which has been removed from the manuscript is blue and has been struck through. We hope these revisions address the concerns of the reviewers.

**Review 1 (submitted 18[th] June 2021)**

*General comments*

*This manuscript presents the impact of updates of the chemistry of biogenic organic compounds and other ozone-related reactions in the UKCA model. At reasonable additional computational cost the chemistry-climate interactions mediated by biogenic organics via ozone and organic aerosols can now be investigated with an up-to-date representation of tropospheric gas-phase chemistry. The manuscript is written very well. The results and the discussion there of is very well structured and comprehensive. With the help of targeted sensitivity simulations the authors back up their explanations for the differences to the results obtained with previous chemical mechanisms used in the UKCA model. Comparison of model results with observational datasets is also very extensive. Interestingly, the model updates result in a significant higher prediction of tropospheric ozone burden exacerbating the positive bias that is typical of other models. Hopefully this gives more stimulus to improve the representation of relevant multiphase chemistry and emissions of precursors in the UKCA model.*

We are pleased to hear that the reviewer finds the paper to be well structured and comprehensive and agree that understanding the model high bias for ozone with CRI-Strat 2 and other models is a priority.

Specific comments

*p2,l54-60: certainly many other studies have investigated these impacts but only two studies from the UKCA are cited. I suggest to cite studies from other modelling communities*

We acknowledge other modellers have conducted research on isoprene and its influence on atmospheric composition and, as suggested, we have included additional references to researchers using the NASA GISS model (Unger et al., 2014), STOCHEM (Khan et al., 2021), ECCHAM5.5-HAM2 (Makonnen et al), TOMCAT (Scott et al., 2015) and an intercomparison of NorESM, ECCHAM and EC-Earth (Sporre et al., 2020). We have also provided a reference to one of the very latest review articles on Isoprene SOA (Claeys and Maenhaut) which highlights the importance of accurately representing isoprene chemistry.

Isoprene's rapid chemical oxidation in the atmosphere by OH, $O_3$ and $NO_3$ directly affects the tropospheric oxidising capacity, ozone burden and the processing of other trace gases like methane (e.g. Archibald et al, 2011, Khan et al., 2021) while also serving as an important source of secondary organic aerosol (SOA) (e.g., Scott et al., 2014, Kelly et al., 2018, Claeys and Maenhaut., 2021). Thus, isoprene has substantial effects on the radiative balance of the atmosphere, both directly via production of SOA and ozone, and indirectly via its changes to the oxidising capacity of the atmosphere influencing methane lifetime and production of other aerosol species such as from oxidation of monoterpenes and $SO_2$ (Unger et al., 2014, Makonnen et al., 2012, Sporre et al., 2020).

*p2,l60: the term "chemical behavior" is borrowed from psychology and just triggers an idriosyncrasy. It is sued in other parts of the manuscript. I would not use it.*

The terms "behaviour" and "chemical behaviour" have been removed from the introduction, Section 4.3.1 and the conclusion with the following amendments:

P2. An accurate representation of isoprene's chemistry chemical behaviour in climate models…

P4. In the CRI framework, species are lumped together into surrogate molecules whose reactivity behaviour is optimised against the fully explicit MCM

P12. While the this "out-of-phase" behaviour nature of the profiles is unlikely to be the sole driver of model-observation difference…

P23. The radiative impact of isoprene, via its influence on atmospheric chemical composition and organic aerosol, means an accurate description of its chemistry chemical behaviour is crucial for

*p4,l114-116: the authors omit the models developed by the groups that discovered and first elucidated the OH-recycling in isoprene chemistry (MAGRITTE in Müller et al., 2019; MOM in Sander et al., 2019 and Novelli et al., 2020).*

Following the reviewer's suggestion, we have added the references to each of the models with the following amended text:

There exist a few reduced mechanisms featuring this state-of-the-art isoprene chemistry suitable for use in chemistry-climate models including the CalTech reduced isoprene scheme (Bates et al., 2019), the MAGRITTE v1.1 model (Müller et al., 2019), the Mainz Organic Mechanism (Sander et al., 2019), the updated ECHAM-MESSy model (Novelli et al., 2020) and the Common Representative Intermediates mechanism v2.2 (CRI v2.2)…

*p14,l499: at this point it is not very clear what isop_ox is made of and that ISOPOOH gives the same PTR-MS signal as MVK and MACR. It would be good it this could be clearly defined.*

We acknowledge the potential for confusion with the two different isop_ox definitions and note that we were restricted by the observational data available. We have amended the text starting from line 492 to clarify the difference between the isoprene oxidation observational datasets and explicitly refer to species considered by using the terms MVK+MACR and MVK+MACR+ISOPOOH.

These terms have been applied throughout Section 4.4 to clarify the situation.

During the GABRIEL flight campaign, the major well known isoprene oxidation products MACR and MVK were measured via PTRMS. These species, along with the ISOPOOH, were also measured at the ATTO tower via PTRMS and are compared with model data. At the ATTO tower, isoprene oxidation products were also measured via PTRMS but in this case were defined as the sum of MACR, MVK and ISOPOOH (Yanez-Serrano et al., 2015) and, to avoid confusion, we refer explicitly to the isoprene oxidation products as either MVK+MACR (for Gabriel) and as MVK+MACR+ISOPOOH (ATTO). In each case, the observational data are compared with model data.

At the ATTO site, all mechanisms are largely high biased for MVK+MACR+ISOPOOH but CS2 produces the best comparison to observations for both diel and vertical profiles (Figs. 1, S9, 11). CS2 also yields the smallest high bias for the ratio of MVK+MACR+ISOPOOH isoprene oxidation products (isop_ox) to isoprene (a metric less sensitive to discrepancies between actual and modelled isoprene emissions) in the Amazon (Figs. 1, S9, 11). Despite the greater oxidising capacity of the PBL in the CS2 simulations, the MVK+MACR+ISOPOOH isop_ox concentrations are lower. This is attributed to the fact that in the relatively low NO$_x$ environment around the ATTO tower, the isomerisation reactions of the isoprene peroxy radical are particularly important and favour the production of HPALDs and other species over MACR, MVK and ISOPOOH.

Relative to the GABRIEL flight data (Fig. 2(d)), the ratio of  MVK+MACR to isoprene is high biased in all mechanisms albeit with the CRI mechanisms exhibiting a smaller bias than ST.

In addition, the captions of Figures 1 and 2 and the titles of the relevant subplots (1(t-y), 2(d)) have been updated to clarify the species involved. The caption and subplot titles in Figures S9 and S11 have also been amended to make it clear that here isoprene oxidation products correspond to MACR+MVK+ISOPOOH, as per the observational of Yanez-Serrano et al (2015).

Figure 1. Mean diel cycles of observed and modelled OH (top row), $O_3$ (2nd row), Isoprene (3rd row) and MT (MT=α-pinene + β-pinene for the CRI mechanisms) (4th row) at the three surface/near surface sites considered. The bottom row shows the vertical profile of the ratio of the isoprene oxidation products MVK+MACR+ISOPOOH to isoprene…

Figure 2. Median observed and model concentrations for the GABRIEL campaign in the Amazon for (a) Isoprene, (b) $O_3$, (c) CO and (d) the ratio of the isoprene oxidation products MACR+MVK to isoprene.

Figure S9 - Mean vertical profiles of the isoprene oxidation products MACR+MVK+ISOPOOH  and the ratio of  MACR+MVK+ISOPOOH to isoprene from observations taken at the ATTO tower (Yanez-Serrano et al., 2015) and model output from ST, CS and CS2. Daytime and nighttime periods are taken as 9:00-15:00 and 21:00-03:00 respectively. Shaded regions indicate ± 1 standard deviation from the mean.

Figure S11. Mean modelled (ST, CS, CS2) and observed diurnal profiles of acetone, MVK+MACR+ISOPOOH isoprene oxidation products  at the ATTO tower. Observations are from the OP3 tower (e.g. Hewitt et al., 2010, Table 3) and model output from the most relevant grid cell. Shading indicates ±1 standard deviation from the observation mean and the numbers in bold show the mean diurnal model bias (model - observations).

*Technical corrections*

*p8,l269: the verb "are" is missing*

The text has been amended as follows:

The emissions used in this study are the same…

**Review 2 (submitted 30th June 2021)**

*This paper presents a comprehensive description of the implementation of the CRI v2.2 tropospheric chemistry mechanism into the UKCA model, and provides detailed comparisons to earlier chemistry versions of UKCA. UKCA is one of the leading global chemistry-climate models, a participant in CMIP6 and widely used by the community, thus it is important to publish documentation such as this. The focus of this work is on the isoprene oxidation updates, and the impacts on the ozone and nitrogen budgets, as well as discussion of potential implications on secondary organic aerosols although the current implementation was not directly coupled to SOA. Appropriate comparisons are made to observations for the evaluation of the chemistry schemes.*

*The paper is well written, and clearly organized. It is quite appropriate for publication in GMD. While the results are specific to the UKCA model the results will be of value to other chemistry-climate model developments and interpretation of their results and limitations.*

*I recommend publication with only minor corrections as noted below.*

We are pleased to hear that the reviewer believes this work to be important and useful for UKCA and other climate models. We have attended to their comments as follows:

*l.273: Should 'new' be 'near'?*
I agree with this suggestion and have made the following change
For the runs done for comparison to observational date recorded at the Z2F site near  Manaus in 2016

*l.346: Section S_? (number missing)*
This typographical error has been corrected with the following change:
… detail in SI Section S3.

*Table S3: SE4C4RS -> SEAC4RS*
This has been corrected in Table S3.

*l.464: Should 'latitude' be 'altitude'?*
We agree that latitude should be altitude and the following change has been made:
… show a low bias as altitude  increases

*l.492: Instead of 'well-known', it would be more appropriate to say 'major' or 'most significant'.*
The substitution of "well-known" with "major" has been made.
..the major  isoprene…

*In general, the PTRMS measurements are reported as the sum of MVK+MACR+ISOPOOH. Please clarify whether or not that is the case here. This paragraph is not clear.*
This point was also raised by Reviewer 1 and clarification of the paragraph has been discussed in the response to Reviewer 1's comment.

*l.579: there are 2 Table S3.*
The "second" Table S2 which refers to the ozone burden and fluxes from the different sensitivity tests has been renumbered as Table S4 and the original Table S4 has been renumbered to Table S5. All references in the text have been updated.

*Section 5.5:*
*Are lightning NO emissions identical in all simulations? Figure 9a looks like it could show a difference in lightning emissions, though I do not doubt it could be explained by the chemistry difference. It would be good to confirm that it was not caused by an inadvertent change in lightning NO.*

This is a good point and I have checked the setup of the model runs for CS and CS2 and confirm that that lightning NOx emissions are the same. As the runs are nudged and lightning NOx emissions are a function of cloud height, they are essentially identical between the runs. This can be seen from the plots below of LiNOx column and zonal mean from the evaluation suites for the CS (cc297) and CS2 (cc298) runs.

**cc297 total column**

[Figure]

**cc297 zonal mean**

[Figure]

**cc298 total column**

[Figure]

**cc298 zonal mean**

[Figure]

**O3 + β-pinene Error**
While working further with the CS2 mechanism, a small error was discovered in the code for two of the reactions of $O_3$ with β-pinene (BPINENE), affecting the runs using the CS2 mechanism. In brief, the rate constants for two of the four reactions of $O_3$ + β-pinene were two orders of magnitude too low which meant that the total rate constant of $O_3$ + β-pinene was ~45% too low.

Reactions with correct rate constants:
BPINENE + O3 = RTX24O2 + OH: 4.73e-16*EXP(-1270/T)
BPINENE + O3 = TXCARB24 + CO: 3.70e-16*EXP(-1270/T)

Reactions with incorrect rate constants and corrections in red:
BPINENE + O3 = HCHO + TXCARB24 + H2O2: 2.7e-18*EXP(-1270/T) → 2.7e-16*EXP(-1270/T)
BPINENE + O3 = HCHO + TXCARB22: 3.38e-18*EXP(-1270/T) → 3.38e-16*EXP(-1270/T)

Fortunately, this does not have a significant impact on general atmospheric composition because, unlike α-pinene, β-pinene's main destruction mechanism is via reaction with OH rather than $O_3$ (due to β-pinene having an exocyclic double bond and α-pinene an endocyclic double bond). Therefore, this correction is an increase of ~45% to a less important β-pinene oxidation pathway.

Nevertheless, to check the influence of this correction we reran two years of longer CS2 simulation (1 year spin, 1 year analysis) and compared it to the first year of analysis from the original CS2 run.

The correction has a negligible effect on tropospheric $O_3$ burden, producing a decrease of 0.07 Tg (0.02%) relative to the original run. Low altitude (<100 m) changes in $O_3$ over the major BVOC emission regions are < 0.4 ppb (< (there is a larger change over the Indian ocean, but this appears to be model noise). Changes in OH over the major BVOC emissions regions are <5%. In both cases, the differences between the original and corrected runs are much smaller than the differences between the CS and CS2 mechanisms.

[Figure]

The tropospheric burden of $\beta$-pinene's decreases by 10% while the burdens of isoprene and $\alpha$-pinene's change negligibly (0.1% and 0.3% respectively). In the lowest 100m $\beta$-pinene's shows decreases up <10% (<0.04 ppb) while isoprene and $\alpha$-pinene show maximum decreases of 2% (<0.01 ppb) and 5% respectively, again much smaller than the difference between CS and CS2.

In terms of reaction fluxes the fraction of $\beta$-pinene reacting with $O_3$ increases from 9.7% to 15.7% with attendant reductions in the fractions reacting with OH (77.2% to 72.7%) and NO3 (13.0% to 11.5%).

The burden of Sec_Org, the tracer used in UKCA to represent oxidised organic species which condense on to aerosol, decreases by 1.2% with the correction. In the lowest 100m, Sec_Org mixing ratios change by <0.5% (<0.01 ppt) over the major emission regions and <3% (<0.05 ppt) over the Boreal forest.

[Figure]

Given the small, localised effect of this change, we decided to rerun only the short CS2 runs used for comparison to the observational datasets which featured high BPINENE concentrations. These are the namely the ATTO, Z2F Brazil, Borneo and Isoprene Columns runs from Table 3.

For all these cases the effect of the correction is very small, as shown by the corrected plots in the marked-up resubmission which closely resemble the originals. Importantly, the effect is much smaller than the differences between the mechanisms. This suggests that the correction will have a negligible effect on the model-observation comparisons for the other sites and so we believe rerunning these is not necessary.

Given the small influence of this correction, we do not believe it is necessary to rerun all the sensitivity tests as it will not materially alter the results, particularly as all of the sensitivity tests had the same $\alpha$-pinene and $\beta$-pinene chemistry as the main CS2 run. We also believe that a complete rerun of the 5-year CS2 run is unnecessary since the only results which have changed by a noticeable amount are the $\beta$-pinene burden and breakdown of the $\beta$-pinene oxidation fluxes. These variables are discussed briefly in Section 5.3, and we have updated this section with the results of the CS2 rerun as follows:

Line 718: However, these differences are dwarfed by the reductions in the burdens of isoprene, $\alpha$-pinene and $\beta$-pinene of 26%, 18% and 24%, respectively.

Line 730: $\alpha$-pinene's chemical destruction by OH, $O_3$ and NO3 changed by 7.5%, -6.3% and -0.8% respectively leading to a total flux increase of 0.05 Tg yr$^{-1}$ (+0.05%). The corresponding changes for $\beta$-pinene with OH, $O_3$ and NO3 were 3.2% , -4.3%  and -0.2%  with a total increase of  0.70 Tg yr$^{-1}$ (1.5%).

Note it was spotted that the original value of a 5.8% increase for $\beta$-pinene + $O_3$ was an error. The new results are in line with the general result of greater OH concentrations from isoprene HO$_x$-recycling. The higher OH concentrations in CS2 lead to a greater fraction of $\beta$-pinene reacting with OH in CS2 than CS (hence 3.2% increase) and a smaller fraction reacting with $O_3$ (-4.3% decrease).

Figures 1,3 and S7-11 have been updated in the main text and the SI. However, this correction does not change any of the arguments made in the manuscript. The rate constants in Section S1 (lines 354-364) are correct.